



# Measurement Report: A comparison of ice-nucleating particle and cloud condensation nuclei sources and properties during autumn at contrasting marine and terrestrial locations

Elise K. Wilbourn[1], Larissa Lacher[2], Carlos Guerrero[1], Hemanth S.K. Vepuri[1], Kristina Höhler[2],
Jens Nadolny[2], Ottmar Möhler[2], and Naruki Hiranuma[1]

[1]West Texas A&M University, Canyon, 79016, U.S.A.

[2]Karlsruhe Institute of Technology, Karlsruhe, Germany

*Correspondence to*: Naruki Hiranuma (nhiranuma@wtamu.edu)

**Abstract**

Ice-nucleating particles (INPs) are an important class of aerosols found worldwide that have far-reaching but poorly quantified climate feedback mechanisms through interaction with clouds and impacts on precipitation. These particles can have highly variable physicochemical properties in the atmosphere, and it is vital to measure their concentration and interconnection with other ambient aerosol populations at a wide variety of sites to comprehensively understand aerosol-cloud interactions in the atmosphere. Toward this aim, we have measured INP concentrations at two contrasting sites, one in the Southern Great Plains (SGP) region of the United States with a substantial terrestrially influenced aerosol population, and one in the Eastern North Atlantic Ocean (ENA) with a primarily marine-influenced aerosol population. These measurements were made at a high time resolution for at least 45 days at each site. From our ENA data, a singular relationship between cloud condensation nuclei (CCN) and INPs was found, suggesting that INPs and CCN originated from the same population of aerosols and potentially the same source. Backward air mass trajectories reveal a strong marine influence at ENA with most air masses originating over the Atlantic Ocean, but analysis of particle chemistry suggested an additional mineral dust INP source at ENA that did not contain large quantities of organic material. This relationship between CCN and INPs was not seen at SGP, and in fact has never been detected before, suggesting that the aerosol particles collected at ENA may represent a unique class of marine aerosols that contain both mineral dust and organics internally mixed in the aerosols and that are capable of acting as both CCN and INPs.



# 1 Introduction

Ice-nucleating particles (INPs) are a proportionally rare population of atmospheric aerosols that assist in the formation of atmospheric ice crystals under ice supersaturation conditions. INPs are present in the Earth's atmosphere in varying concentrations, ranging from $10^{-6}$ to $10^{3}$ L$^{-1}$ over wide freezing temperatures, and they come from both anthropogenic (e.g., manufacturing, transportation, soot, biomass burning, and agriculture) and natural (e.g., maritime, terrestrial bacteria, volcanic sulfate, biomass burning, K-feldspar/mineral, and soil dust) sources (Kanji et al., 2017). INPs supply surfaces for the deposition and freezing of water vapor and/or cloud droplets, leading to a type of ice formation known as heterogeneous freezing. In contrast, homogeneous freezing, which requires cloud droplets to be cooled to a temperature of approximately -32 °C or below, occurs in the absence of INPs (Koop and Murray, 2016; Koop et al., 2000).

Ice formation in climate models is a source of uncertainty in current models, warranting further study of ice nucleation processes and INPs (Knopf and Alpert, 2023; Burrows et al., 2022; Forster et al., 2021; Murray et al., 2021). While understanding of INPs remains limited, recent advancements have shed light on the various modes of heterogeneous ice-nucleation in the atmosphere. INPs can alter the altitude of ice cloud formation and influence nucleation and freezing pathways. For instance, through immersion freezing processes, which predominate over 85% of atmospheric heterogeneous freezing, water droplets containing these aerosol particles freeze at temperatures higher than would be possible with pure water alone (Hande and Hoose, 2017; Westbrook and Illingworth, 2011). Other nucleation pathways include deposition nucleation (and/or pore condensation freezing) (David et al., 2019; Marcolli, 2014), condensation nucleation (Hoose and Möhler, 2012), and contact nucleation ( Fornea et al., 2009; Durant and Shaw, 2005). Secondary ice formation processes may also lead to an increase in atmospheric ice crystals (e.g., Korolev and Leisner, 2020; Sullivan et al., 2018; Field et al., 2017) but will not be addressed in this study.

Cloud condensation nuclei (CCN) are an additional subset of aerosols responsible for changing cloud properties, including radiative effects, lifetime, and precipitation processes. These particles are hygroscopic, leading to the formation of cloud droplets when the vapor pressure of water is raised above saturation (Petters and Kreidenweis, 2007; Köhler, 1936). Important predictors of a particle's ability to act as a CCN are its hygroscopicity and size (Burrows et al.,





2022; Dusek et al., 2006). An increase in the hygroscopicity of particles enables them to act as CCN at lower water supersaturations (Lohmann et al., 2016), although organic aerosols are not necessarily hygroscopic but still a major source of CCN ( Cruz and Pandis, 1997; Novakov and

Penner, 1993). Some of the most studied and best CCNs are salts and particles containing salts, including ammonium sulfate and sodium chloride (Schmale et al., 2018). CCN can be present in local aerosols or transported to the region of measurement, but another known subset of CCN forms via the growth of particles within boundary layer conditions (Spracklen et al., 2008). As ice nucleation can involve first the formation of a droplet around the INP, it is possible for a particle

to act as both a CCN and then an INP within the same droplet.

Marine sea spray aerosols (SSAs), produced during jet spray and wave breaking (Wang et al., 2017), were found to contain both CCN and INPs (Brooks and Thornton, 2018; Wilson et al., 2015; Andreae and Rosenfeld, 2008). The bubble-bursting process aerosolizes the organic material found in the underlying seawater and particularly the material found in the sea surface microlayer,

forming organic- and salt-rich SSA. SSAs containing marine organic material are well-known and globally assumed to be a potential source of INPs (Burrows et al., 2013). SSAs dominate the aerosol populations at sites with strong marine influences and few anthropogenic inputs, including the first study site addressed here. This site, a United States Department of Energy (DOE) Atmospheric Radiation Measurement (ARM) site in the Azores on Graciosa Island will be referred

to throughout as the Eastern North Atlantic (ENA) site.

Most previous studies have not found a relationship between the population of aerosols that act as INPs and those that act as CCNs ( Gong et al., 2020a; Gong et al., 2020b). Studies of simultaneous marine CCN and INP measurements made at high time resolution are severely lacking, with only a few studies undertaking this (Gong et al., 2019; Gong et al., 2020a; Gong et

al., 2020b). However, Roesch et al. (2021) demonstrated that at specific supersaturation conditions, mineral dust is capable of acting both as a CCN and as an INP. Additionally, secondary marine aerosols that have undergone chemical oxidation processes are known to act as CCN, potentially at a higher rate of importance for cloud formation than primary marine aerosols (Mayer et al., 2020). Zheng et al. (2018) conducted a long-term (3-year) study of CCN and other aerosols

at ENA and found that many of the CCN at ENA can be attributed to particles entrained into the marine boundary layer from the free troposphere that then grew into CCN-active particles.





There are few INP measurements from the temperate oligotrophic Atlantic Ocean and only one at ENA, leaving a potential knowledge gap and increasing model uncertainty in the region. The only prior offline INP measurements were taken for the samples collected in the Azores as part of the Aerosol and Cloud Experiments in the Eastern North Atlantic (ACE-ENA) study in 2017 and 2018 (Knopf et al., 2022), but these studies were regarding shorter intensive operating periods. However, there are several studies on both aerosol properties (Zheng et al., 2022; Wang et al., 2021a; Zawadowicz et al., 2021; Gallo et al., 2020; Zheng et al., 2018), air mass origins (Wang et al., 2020; Véron and Church, 1997), and CCN (Wood et al., 2017) at ENA. These studies indicate a site with strong marine influence with the majority of aerosols classified as boundary layer SSAs (Wang et al., 2021b), with some also including continental dust and/or anthropogenic aerosols (Véron and Church, 1997).

To contrast with the primarily marine nature of the ENA site, here we compare results from the Southern Great Plains (SGP) site in Oklahoma, US (also a US DOE ARM site) as an example of a site with strongly terrestrial influences and little local anthropogenic influence on aerosols. Two prior campaigns have addressed INP concentrations, $n_{INP}$, at SGP (Knopf et al., 2021; Demott et al., 2015), and several others have addressed other aerosol properties at SGP without regard to INPs ( Liu et al., 2021; Fast et al., 2019; Logan et al., 2018; Sisterson et al., 2016; Parworth et al., 2015; Wang et al., 2006). Demott et al. (2015) measured $n_{INP}$ with a continuous flow diffusion chamber in May and June and found dust and long-range biomass-burning material in the aerosols present at the site, presumably contributing to INPs at least in part. In addition, the agricultural fields surrounding the sampling site are a source of organic material and likely also act as INPs in aerosol particle samples gathered at SGP. Knopf et al. (2021) found organic carbon in all particles examined as part of the study. Knopf et al. (2021) also measured $n_{INP}$ using CFDC and a Portable Ice Nucleation Experiment chamber (PINE), a small-scale 10 L vessel expansion chamber described in Möhler et al. (2021), as *in-situ* INP monitors, over a single day to better incorporate $n_{INP}$ into climate models.

This study reports the measurement data from two field campaigns, including Examing the Ice-Nucleating Particles from SGP (ExINP-SGP; https://www.arm.gov/research/campaigns/sgp2019exinpsgp) and another one from ENA (ExINP-ENA; https://armweb0-stg.ornl.gov/research/campaigns/ena2020exinpena), aims to quantify INPs and CCN measured at the two DOE-ARM sites, one marine-influenced and one with terrestrial



and little marine influence on aerosols, and determine the potential relationship between CCN and INPs at the two sites. Additionally, the physicochemical properties of bulk aerosol particles

sampled at the site were examined with the aim to both further describe the source of the two populations and to determine whether the two types of aerosols share a source. Due to the distinctiveness of the ENA measurement site and the lack of comparable studies, the same methods were employed at SGP as a continental measurement site. The high-time resolution data analyzed here provide first-of-its-kind information about CCN and INPs that could be incorporated into

global climate models to reduce the uncertainties associated with current aerosol measurements (Murray et al., 2021).

## 2  Methods
### 2.1. Study Sites

Online measurements and sampling activities were conducted at two field sites operated by the DOE ARM program. The locations of the two sampling sites are shown in Figure 1. The first sampling site, the ENA site, is located on Graciosa Island in the Azores at 39° 5' 29.76" N, 28° 1' 32.52" W (Wood et al., 2015). While measurements were made at ENA from October 1, 2020, to March 28, 2021 (UTC), the analysis here will focus on sampling from the autumn period from

October 1 to November 30, 2020, UTC). Although ENA is 1500 km from the nearest continental land mass, Saharan dust has been observed at the site (Logan et al., 2014). Entrainment and transport of large quantities of Saharan Dust by the Azores High is the primary method of transportation of dust across the North Atlantic Ocean (Doherty et al., 2008). It should be noted that the center of the Azores High is not necessarily centered over the Azores themselves but is

known to vary (Mächel et al., 1998). However, the ENA site does show a strong influence from the Azores High (Rémillard and Tselioudis, 2015), which can entrain Saharan Dust and bring this material to ENA.

Although there are small towns on islands in the Azores, most of the influence at the ENA site is marine (Wood et al., 2015). Possible anthropogenic influence comes primarily from

transportation, as the sampling site is near the island's airport to the north and the road to the south that has large firetrucks traveling down it multiple times daily on the way to the airport. There are generally two to three flights arriving and departing to and from Graciosa each day, but the



schedule was not consistent across the sampling period, so anthropogenic influences were determined from black carbon levels using methods described in Section 2.5.2.

The SGP site is located in Oklahoma in the central United States at 36° 36′ 26.36″ N, 97° 29′ 15.51″ W. This site is surrounded by farmland, with the nearest city dozens of kilometers away (Sisterson et al., 2016). Sampling activities at the SGP site took place from October 1 to November 15, 2019 (UTC). The SGP site is the oldest DOE ARM site, established in 1992 as the first Clouds and Radiation Testbed (Stokes and Schwartz, 1994). The nearest large water body is the Gulf of

Mexico, and any marine aerosols would be transported and mixed with continental aerosols. The site has distinct seasons and variable wind sources (Sisterson et al., 2016; Stokes and Schwartz, 1994). More than a dozen instruments are present at both sites and collect continuous data, with many of the same instrument models between both sites. An exact list of instrumentation and models can be found in Table S1 from supplemental information, SI, Section S1.


## 2.2. Aerosol concentrations ($n_{aer}$) and ambient conditions

Aerosol concentrations ($n_{aer}$) were measured at both sites with condensation particle counters (TSI, Inc. model 3772). Ambient meteorological conditions, including wind speed, wind vector direction, temperature, pressure, relative humidity, and precipitation rates, were measured at both

sites with Vaisala weather transmitters (Model WXT520). All instruments were associated with the ARM Aerosol Observing System (other than INP measurement instruments including a filter-based aerosol particle sampling system for offline INP analysis). To compare with data collected at longer time scales, all online data sets were averaged over six-hour time periods.

**2.3. Ice-Nucleating particle concentration ($n_{INP}(T)$) measurements**

At both the ENA and SGP sites, the same instrumentation was used for online measurements of $n_{INP}$ as a function of temperature ($n_{INP}(T)$, L$^{-1}$ air) and aerosol particle sampling activities. Online INP measurements and aerosol particle sampling activities for offline INP measurements were made concurrently, although the time resolution, as well as the examined freezing temperature

range, was different. Aerosol particles were collected through similar quasi-laminar flow, 5.5 m height inlets constructed with aluminum pipes at the sampling sites (4-inch and 6-inch diameter pipes at ENA and SGP respectively). Inlets were capped with a Total Suspended Particle (TSP)





inlet and samples were drawn through the inlet into an air-conditioned sampling trailer. The stack inlet was topped with a total suspended particle inlet.


### 2.3.1. Portable Ice Nucleation Experiment Chamber

The $n_{INP}$ values were measured at both sites with the same PINE version 3 (Bilfinger Noell GmbH, PINE-3 hereafter). PINE-3 is capable of measuring but not distinguishing between both immersion mode and deposition mode freezing events. PINE-3 measurements were carried out through a quasi-laminar stack inlet distinct from the inlet for all other aerosol measurements. At ENA, the stack inlet was connected to PINE via a 3/8 inch internal diameter copper tube. Sampling to PINE at SGP was conducted through a similar quasi-laminar stack inlet (4-inch diameter, 5.5 m height) connected to a 3/8 inch internal diameter copper tube that ran from the stack inlet to the PINE-3. At SGP, these activities were carried out at the ARM guest instrument facility. At ENA, a dedicated air-conditioned trailer was used for online sampling activities. More information on the inlets and inlet particle loss can be found in SI Section S2.

PINE operates by cycles of flushing ambient dried air through a cooled chamber and subsequently forced expansions of the sampled air within this chamber. During the expansions, the sample gas temperature and the pressure are constantly lowered such that supersaturated conditions with respect to ice and liquid water are created, to induce ice nucleation in the presence of INPs. Particles exiting the chamber pass through an optical particle counter (OPC; fidas-pine; Palas GmbH), and ice crystals are differentiated from smaller aerosol particles and/or water droplets on the basis of their optical size. The air gas temperature in the chamber was changed between -10 °C and -30 °C at SGP, and -14 °C to -33°C at ENA. The time resolution of such a temperature cycle was approximately two hours. At SGP, the chamber was flushed with ambient air through a 5.5 m laminar flow stack inlet (see SI Section S2) for 300 seconds at a 2 standard liter per minute (lpm) of air flow rate, followed by an expansion with 3 lpm of pump flow to a 750 mb internal pressure and a refill with ambient air at 2 lpm back to ambient pressure. At ENA, the chamber was flushed at 2 lpm for 600 seconds, followed by expansion with 3 lpm of pump flow to an 800 mb internal pressure, and then refilled at 2 lpm to ambient pressure. With the flushing conditions at both ENA and SGP, we ensured to measure the replaced population of aerosol particles in the vessel in each run. PINE was also cleaned with a daily cleaning cycle where filtered ambient dry air was flushed through the chamber until no particles were detected anymore in the





OPC during the expansion mode. The calibration of PINE-3 is described in SI Section S3.

Moreover, the instrument was defrosted approximately once per month using methods described in sections S4 and S5. A leak test (as described in Section S6) was performed several times during the sampling period, and a vibration test (Section S7) was performed at ENA. This ensures that no internal ice crystals formed and could have led to a high-bias in $n_{INP}$.

The time resolution of one expansion at SGP is approximately 8 minutes, and

approximately 12 minutes at ENA due to the longer flush time. The $n_{INP}$ values normalized to a unit volume of sampled air were calculated from PINE-3 data following the method described in Möhler et al. (2021). The concentration of INPs measured with PINE-3 ($n_{INP}$, standard L$^{-1}$ air) is calculated using the following equation:

$$n_{INP}(T) = \frac{\Delta N_{ice}}{F_{em}\Delta t_{em}} = \frac{\Delta N_{ice}}{V_{em}} \qquad [1]$$

where $\Delta N_{ice}$ is the count of ice crystals measured with the OPC, $F_{em}$ is the volume of expansion ($V_{em}$) divided by the duration of the expansion, and $\Delta t_{em}$ is the duration of the expansion. More detail on PINE-3 data analysis can be found in SI Sections S8 and S9, and information on the systematic and statistical error inherent in PINE measurements can be found in Section S10. Please see Möhler et al. (2021) for more information about PINE. It is noteworthy that the minimum

temperature measured during the expansion represents the temperature for $n_{INP}$ for each run in this study.

### 2.3.2.  Collection of Aerosol Particles on Filters for Offline Analysis

Filters were collected at both ENA and SGP using a filter impactor to sample onto cleaned 47 mm

polycarbonate filters (Whatman Nuclepore, 0.2 µm pore size). For cleaning, filters were soaked in 0.05 vol% hydrogen peroxide and dried at room temperature prior to sample collection to remove any preexisting organic contaminants, and blank filters were also periodically collected at each sampling site. Filters were collected for approximately 4 days at around 7-10 lpm (see SI Section S11 for exact filter time periods and airflow information). Filters were stored in sealed, sterile petri

dishes at -20 °C prior to analysis (other than during sampling and transportation, which took up to 14 days in total), which occurred no more than 12 months after collection.

Filters from ENA were analyzed using the West Texas Cryogenic Refrigerator Applied to freezing Test (WT-CRAFT) instrument (Vepuri et al., 2021). Taking into account the expected



particle concentration, filters were placed in a calculated volume of HPLC-grade water (Sigma Aldrich) in sterile tubes (15 mL, VWR), shaken for five minutes to liberate particles, and allowed to stand for one minute. Samples were placed onto an aluminum plate coated with clear petroleum jelly (70 droplets, 3 µL volume each) and placed into the cryocooler. A video was recorded as the samples were cooled at a rate of 1 °C per minute and analyzed at 0.5 °C increments to determine the fraction of droplets frozen. A complete dataset includes data from 0 °C to -25 °C. Although the cryocooler is capable of reaching temperatures below -25 °C, using data from this range ensures that we only include data that can be attributed to immersion-mode freezing events without artifacts (see SI Section S11). Samples were diluted with HPLC-grade water 10 times or 100 times as needed, and the lowest calculated $n_{INP}$ was reported at each 0.5 °C increment to prevent overestimation of $n_{INP}$ (Vepuri et al., 2021). Our HPLC-grade water is virtually INP-free at -25 °C (Wilbourn et al., 2023). Data from WT-CRAFT has a minimum INP detection limit of 0.001 L$^{-1}$. The temperature uncertainty for WT-CRAFT is ± 0.5 °C with a 23.5% uncertainty in $n_{INP}$ (Vepuri et al., 2021). A 95% confidence interval was calculated for each 0.5 °C data point as described in Schiebel (2017), and the analysis of blank filters is discussed in SI Section S11.

Filters from SGP were analyzed with the Ice Nucleation Spectrometer of the Karlsruhe Institute of Technology (INSEKT) system (Schneider et al., 2021; Schiebel, 2017). Samples on filters are resuspended in 8 mL of filtered nano-pure water and mixed. This volume of water was based on the volume of air filtered through each filter and ensured that the minimum detection limit was 0.015 INP L$^{-1}$. Samples are then aliquoted into wells in a polystyrene polymerase chain reaction (PCR) plate (50 µL per well) that is cooled at 0.33 °C per minute. Sample freezing is observed through a camera based on light transmission intensity. The temperature uncertainty associated with INSEKT is ± 0.5 °C. Both offline instruments are employed to detect freezing events at temperatures warmer than PINE-3. WT-CRAFT and INSEKT provide equivalent data and have been compared previously (Hiranuma et al., 2021).

Samples collected on filters were also treated with 100 °C heat to remove heat-sensitive material, including but not limited to proteins, which denature at temperatures above approximately 60 °C (Hogg, 2013). A 1 mL portion of the suspension containing the sample was placed into a sterile 15 mL polycarbonate tube (VWR), which was then capped and placed into a beaker of boiling water for 20 minutes. The sample was allowed to cool and then $n_{INP}$ was



measured with either WT-CRAFT or INSEKT. The heat-treated sample was also diluted as needed

to collect data between the warmest freezing point and -25 °C.

The number of INPs ($n_{INP}(T)$, $L^{-1}$ air) collected on a filter sample can be calculated as

$$n_{INP}(T) = -\ln\left(\frac{f_{unfrozen}(T)}{V_{drop}*10^{-6}}\right) * \frac{dilution\ factor}{\left(\frac{V_{air}}{V_{water}*10^{-3}}\right)} \qquad [2]$$

where $f_{unfrozen}$ is the fraction of droplets unfrozen at a given temperature, $V_{drop}$ is the individual

droplet volume (3 µL for WT-CRAFT, 50 µL for INSEKT), $V_{air}$ is the volume of air sampled (L),

and $V_{water}$ is the volume of HPLC-grade nano pure water used to resuspend the sample (mL, see

Tables S5 and S6 from SI Section S11 for amounts used for each filter). The number of aliquots

made for each method was chosen to increase statistical validity.

For samples analyzed with WT-CRAFT, if sample data at full strength did not reach the

minimum temperature (-25 °C) the sample was diluted by 10 times with HPLC-grade water and

the diluted sample $n_{INP}$ was measured using the same method. Samples analyzed with INSEKT

were diluted 15 and 225 times. To prevent over-estimation of $n_{INP}$, if there was an overlap between

the diluted and full-strength measurements (for data from both WT-CRAFT and INSEKT), the

lower of the two calculated $n_{INP}$ at each 0.5 °C was chosen (note the methodological difference

from other studies using INSEKT data, which use the data point with the lower uncertainty). If the

diluted profile did not match the full-strength INP profile within the 95[th] percentile range, then the

diluted sample was re-run.

### 2.4. Cloud Condensation Nuclei Concentration ($n_{CCN}$(%SS)) Measurements

CCN concentrations ($n_{CCN}$(%SS)) at ENA and SGP were made with CCN counters (ENA: CCN-

100, SGP: CCN-200, both from Droplet Measurement Technologies) (Uin, 2022; Jefferson, 2011;

Roberts and Nenes, 2005). All measurements at SGP were made with the CCN-200 using one

column to scan through supersaturations from 0.1% to 1.0% and one column measured at a fixed

supersaturation of 0.4%. Measurements made at ENA with the CCN-100 were scanned through

seven supersaturations from 0.1% to 1.0%. The uncertainty of activated optical particle sizing for

the ARM CCN counters is reported to be ± 0.25 µm, and the accuracy of supersaturation based on

the accuracy of pressure, flow, and temperature sensors is estimated to be ± 3% (Uin, 2022) Data

is only reported at 0.1 and 0.2% supersaturation in this study as natural systems rarely reach higher

supersaturations. The lower bound (i.e., 0.1%) was set because the supersaturation dependence on



the humidifier column temperature gradient becomes non-linear below this threshold supersaturation and the associated accuracy can be as low as 40%. A full scan took approximately an hour for both instruments to allow the instrument time to stabilize at each measured supersaturation. We note that the ENA-ARM CCN data were available for October 2$^{nd}$ to 26$^{th}$, 2020, while the SGP-ARM CCN data were available for the entire ExINP-SGP campaign period (October 1$^{st}$ to November 15$^{th}$, 2019).


## 2.5. Aerosol Chemistry Measurements

### 2.5.1. Aerosol Chemical Speciation Monitor

Bulk aerosol particle chemistry at ENA and SGP was studied using an Aerosol Chemical Speciation Monitor (ACSM, Aerodyne Inc.). This instrument is based on the Aerodyne Aerosol
Mass Spectrometer (Canagaratna et al., 2007; Jayne et al., 2000) but has been modified for long-term deployment to limit the down time needed for recalibration and instrument maintenance (Ng et al., 2011). The ACSM gives information on non-refractory aerosol particles and was used to determine the concentrations of aerosol particles in the following categories: sulfates, nitrates, ammonium, total organics, and chloride. Bulk chemical composition data were available for the
entire sampling period at SGP, and from November 14, 2020, to November 30, 2020, at ENA. The time resolution of ACSM data used is 30 min for signal averaging (Watson, 2017).

### 2.5.2. Black Carbon Mass Concentration ($m_{BC}$)

Anthropogenic influence can be seen in high $m_{BC}$, often from local biomass-burning or
transportation vehicles. Although $m_{BC}$ was not measured directly, it can be calculated based on light transmission measured by a Particle Soot Absorption Photometer (PSAP, Radiance Research) (Springston, 2018). PSAP data were corrected according to Bond et al. (1999) and Ogren (2010) to account for filter loading over time. PSAP instruments operate on the principle that aerosol absorbance is dependent on particle composition, with the absorption at 529 nm most
representative of black carbon aerosols. This assumption of uniform aerosol composition may introduce uncertainties in information derived from PSAP data, as few natural aerosol populations have uniform composition. The mass of black carbon was estimated by dividing the absorption at





529 nm by the estimated mass absorbing cross section of 7.5 m$^2$ g$^{-1}$ (Zheng et al., 2018; Bond et al., 2013) to obtain an estimate of the mass of black carbon present in aerosols ($m_{BC}$, ng m$^{-3}$).


### 2.6. Air Mass Origin and Statistical Methods

#### 2.6.1. Back Trajectory Analysis and Geographic Classification of Air Mass Origins

Backward air mass trajectories were calculated using the Hybrid Single-Particle Lagrangian Integrated Trajectory (HYSPLIT) model (Rolph et al., 2017; Stein et al., 2015) (available at

https://www.ready.noaa.gov/HYSPLIT.php) to compute archive trajectories every six hours during the sampling period. Each 240-hour backward trajectory was calculated at three heights: the height of the sampling inlet (5 m AGL), the planetary boundary layer height, and the cloud base height. While the impact of cloud scavenging on aerosol particles, as well as dry and wet depositon, is not considered in this study, at least the air mass trajectory analysis at multiple heights

is offered. The planetary boundary layer height defines the amount of the troposphere in direct interaction with the ground surface, while the cloud base height was the height of the first cloud layer. Both the planetary boundary layer height and the cloud base height were calculated using data from a ceilometer (Model CL31, Vaisala) (Morris, 2016). Further detail on the calculation of the planetary boundary layer height can be found in Sivaraman et al. (2013). The origin of the back

trajectory was classified into broad regional categories, including the major oceans and continents. More information on the air mass origin classifications for ENA and SGP can be found in SI Section S12. As precipitation removes aerosols via wet scavenging, the amount of precipitation was examined along each back trajectory. Starting at ENA and moving backwards along the trajectory, the trajectory was cut off at the point where the sum of all rainfall reached 7 mm and

the origin was reclassified as needed if the new origin was less than 240 hours prior (Gong et al., 2020a).

#### 2.6.2. Statistics

Spearman's rank-order correlations were used to determine the strength of relationships between

variables. A relationship was described as strong with a $\rho$ value greater than 0.7, moderate with $\rho$ between 0.5 and 0.7, and weak between 0.3 and 0.5 (all p-values <0.05). Pearson product-moment



correlations were calculated and a relationship was described as strong with an r value greater than 0.7, moderate with r between 0.5 and 0.7, and weak between 0.3 and 0.5 (all p-values <0.05).

## 3 Results

### 3.1. Ambient Atmospheric and Aerosol Conditions

The ambient measurements taken at the two sites show very different conditions. Detailed numbers can be seen in section S13. The temperature at SGP was lower, with an average temperature of 10.2 °C compared to 18.3 °C at ENA, even though the ENA study period continued further into 370 the autumn season. The mean relative humidity at ENA was higher, at 76.4% compared with 65.5% at SGP. Although the Great Plains are known to have strong winds, the average windspeed at the two sites was comparable, with an average windspeed of 5.3 m s$^{-1}$ at ENA and 5.1 m s$^{-1}$ at SGP. In fact, the maximum windspeed at ENA of 11.4 m s$^{-1}$ was slightly higher than the maximum windspeed of 10.4 m s$^{-1}$ at SGP. The wind at both sites predominately came from the south, 375 although both sites had variability in wind direction and air mass origin that will be addressed in Section 3.4.

Figure 2 shows the total $n_{aer}$ measured using the same model CPC at both locations. The median $n_{aer}$ (± standard error) at ENA was almost an order of magnitude lower than at SGP at 393.25 ± 30.85 cm$^{-3}$ compared with SGP at 3055.00 ± 87.83 cm$^{-3}$. Although there were times 380 when $n_{aer}$ at SGP was well below 500 cm$^{-3}$, the concentration of particles was much more variable and on average higher at SGP than ENA. The maximum $n_{aer}$ at SGP was also higher (5677.39 cm$^{-3}$ compared with 3427.59 cm$^{-3}$ at ENA). This difference in concentration can primarily be attributed to the difference between continental and marine sampling sites. Due to their closer proximity to aerosol sources, continental sites generally have higher total aerosol mass and number 385 concentrations than marine sites. The dominant aerosol sources at marine sites are generally limited to transported material and new particle formation through marine boundary layer interaction with the free troposphere (Clarke et al., 2013; Katoshevski et al., 1999; Russell et al., 1994) and the generation of sea spray aerosols through wave breaking and bubble bursting (Fuentes et al., 2010; Christiansen et al., 2019; Cochran et al., 2017). Transported aerosols at ENA 390 have been observed from sources as distant as North America (Zheng et al., 2020; O'Dowd and Smith, 1993) based on air mass backward trajectories, and will be further addressed in Section 3.4.





Although the average $n_{aer}$ differed by an order of magnitude between the two sites, estimated $m_{BC}$ (plotted with blue crosses in Figure 2) are much closer, with an average $m_{BC}$ of 0.59 ± 0.08 ng m$^{-3}$ at ENA and 0.74 ± 0.06 ng m$^{-3}$ at SGP (average ± standard error). The maximum

$m_{BC}$ was also nearly four times higher at ENA when compared with the maximum $m_{BC}$ at SGP (13.67 ng m$^{-3}$ and 3.35 ng m$^{-3}$, respectively), and the spread of $m_{BC}$ values is slightly greater at ENA than at SGP. This is likely due to local anthropogenic black carbon sources at ENA, which are within 1 km of the airport and major road. There was no relationship between wind direction and $m_{BC}$, suggesting that this local source does not dominate the total black carbon present at the

site. However, it should be noted that transported biomass-burning material has been previously observed at the ENA site (Wang et al., 2021a; Zheng et al., 2020), so a portion of the ENA black carbon may be from material transported from North America, especially during time periods with North American origin for local air masses as indicated by back trajectories. This is confirmed by a moderate relationship between black carbon mass and $n_{aer}$ ($\rho = 0.56$, p < 0.05) at ENA. At SGP,

less of the black carbon is due to local anthropogenic production; instead, it is dominated by transported black carbon from biomass-burning and wildfires (Logan et al., 2018). There is no statistically significant correlation between $n_{aer}$ and $m_{BC}$ at SGP ($\rho = -0.11$, p = >0.05), suggesting that the aerosol population is not driven by black carbon and the majority of particles are dominated by material other than black carbon.


### 3.2. Online Ice-Nucleating Particle Concentrations

Shown in Figure 3 is the comparison of $n_{INP}(T)$ from two sites for similar freezing temperature and measurement time ranges. The time series of PINE-3 6-hour averaged $n_{INP}(T)$ from SGP for freezing temperatures from -15 °C to -30 °C with a temperature resolution of 1 °C is shown in

Figure 3A, with the color of each point corresponding to the freezing temperature. The freezing temperature (operationally defined) was lower at SGP, but both measurement periods included heterogeneous INP measurements at temperatures as low as -30 °C. It should be noted that the apparent lack of low-temperature INPs at SGP in the middle of the sampling period is due to the intended measurements at above -20 °C. However, there was a measured decrease in INPs active

at temperatures below -25 °C just prior, from October 25-27, 2019. As summarized in the table in SI Section S13, low-temperature INPs measured at ENA during the same time of year show less variability, with a maximum 6-hour average was $n_{INP}$(-30 °C) of 161 L$^{-1}$. For SGP, a maximum 6-



hour averaged was 286 L$^{-1}$. There were measurements at both locations where the $n_{INP}(T)$ was below the detection limit of 0.3 INP L$^{-1}$, making it difficult to determine the minimum $n_{INP}(T)$ at

each location.

Figure 3B displays the time series of PINE-3 6-hour averaged $n_{INP}(T)$ from ENA for freezing temperatures from -20 °C to -30 °C with a temperature resolution of 1 °C. Although the measurements at SGP were made in 2019 and the measurements at ENA were made in 2020 (Figure 3B), the patterns in $n_{INP}(T)$ can still be compared as representative of the entire autumn

season, although an ideal dataset would include multiple years to remove any bias that might be introduced from a single year. When comparing the median $n_{INP}(T)$ from October – November for both locations (also refer to Table S7), it is apparent that $n_{INP}(T)$ at SGP is consistently higher than $n_{INP}(T)$ at ENA at all measured temperatures. Although the measured $n_{INP}(T)$ at both sites are closer at temperatures between -22 °C and -20 °C, there is consistently around an order of magnitude

difference in $n_{INP}(T)$ between the two sites, which can be in part attributed to the difference in median $n_{aer}$ (i.e., 393 cm$^{-3}$ and 3055 cm$^{-3}$ for ENA and SGP, respectively). As well, the data from SGP can be compared with previously collected $n_{INP}(T)$ data as reported by DeMott et al. (2015) for April-June 2014 and Knopf et al. (2021) for October 2019, where it becomes apparent that the data collected in this study is comparable to data collected in the spring to early summer season

2014 and measurements of $n_{INP}(T)$ made by other online instrumentation in Fall 2019.

Although there are no previous studies reporting long-term $n_{INP}(T)$ at ENA, in Figure 4 we have compared the data to all other studies reporting $n_{INP}(T)$ in locations with strong Atlantic Ocean influence (Wilbourn et al., 2023; Figure S1 and references therein). The $n_{INP}(T)$ range for these studies is much larger than the range reported at SGP, potentially due to either difference in

sample techniques or differences in $n_{INP}(T)$ between the seven locations. By comparing previous studies to our current ENA data, we can conclude that our data falls within the range of $n_{INP}(T)$ reported by them.

As the data reported here are from a single season, no seasonal cycles can be elucidated. However, at ENA a decrease in $n_{INP}(T)$ at all measured activation temperatures over the course of

the autumn was found ($\rho$ = -0.25 to -0.64, p < 0.05 in Table S8). The decrease seen at ENA, although statistically significant, was minor and not noticeable by visually assessing our plots. Regardless, this decrease may be due to a decrease in the aerosol particle counts measured by CPC (while it is not statistically significant; i.e., $\rho$ = -0.11 in Table S8) and/or Northern Hemisphere

 

biological activity, which can impact biogenic aerosol generation and emission. This decreasing
trend of the CPC counts is not statistically as significant as what is seen in $n_{INP}$. Therefore, the
decrease in the aerosol particle counts itself would not solely explain the $n_{INP}$ decreasing trend over
time. Mineral dust is a known source of INPs at both ENA and SGP, but the mineral dust
concentrations do not follow seasonal patterns at ENA or SGP according to previous studies
(Knopf et al., 2022; Knopf et al., 2021; DeMott et al., 2015) suggesting the importance of non-
mineral dust INPs contributing to the seasonal variability in $n_{INP}(T)$.

Figure 5 shows the 6-hour average $n_{INP}(T)$, as well as INP activated fraction ($IAF =$
$n_{INP}(T)/n_{aer}$) at selected temperatures (-15 °C, -20 °C, -25 °C, and -30 °C). Despite the substantial
difference in $n_{INP}$ ($n_{INP,SGP} >> n_{INP,ENA}$ seen in panels A and B), it appears that $IAFs$ at -20 °C and
-25 °C are very similar for ENA and SGP ($\approx$ in the order of $10^{-5}$ to $10^{-6}$; see panels C and D).
Moreover, $IAF$ interestingly tends to be higher at -30 °C at ENA with a median $IAF$(-30 °C) of
approximately 4.6 x $10^{-5}$ than SGP ($\approx 1.7$ x $10^{-5}$) while we cautiously note that high variability in
$n_{INP}$(-30 °C) and the aforementioned, intended high temperature measurements at SGP may play a
role in this trend. Nonetheless, this $IAF$ trend suggests that (1) $n_{INP}(T)$ does not necessarily scale
to $n_{aer}$ (and vice versa) and (2) aerosol population in ENA was found to generate more INPs active
at low temperatures, which represents a unique finding of this study. Additionally, the relationship
between $n_{INP}(T)$ measured by PINE-3 and other online measured variables during ExINP-ENA
and -SGP is discussed in SI S14. It should be noted that this study focuses on reporting $n_{INP}$ from
distinctly different locations, and the discussion of freezing efficiency (i.e., $n_{INP}$ normalized to
aerosol particle properties) is beyond the scope of this measurement report. Detailed and focused
studies investigating the relationship between INP and particle size distribution, as well as the
composition of key relevant species, are available elsewhere (e.g., Knopf et al., 2021; Raman et
al., in preparation).

**3.3. Offline Ice-Nucleating Particle Measurements and INP Heat Sensitivity**

The $n_{INP}(T)$ measured from filters gives values at temperatures higher than the operating
temperature of PINE-3 due to the larger sample volume. At the same time, the temperature range
also overlaps with the range above -25 °C measured with PINE-3, allowing for comparison and
validation of the two techniques.



Figure 6 summarizes the results of offline INP measurements and associated heat-treated INP experiments from the two sites. As shown in Table S7, the aerosol particle samples collected at ENA (N = 18) showed a lower $n_{INP}(T)$ at all temperatures when compared with SGP (N = 21), confirming the pattern seen with PINE-3 measurements. As inferred from Figure 6, the average initial freezing temperature (± standard deviation) of samples collected at ENA (-12.4 ± 3.4 °C) was also lower than that of samples collected at SGP (-6.4 ± 0.7 °C), suggesting the terrestrial INPs active at high temperatures at SGP were not present at ENA. At SGP, the average $n_{INP}$(-10 °C) was approximately $10^{-1}$ INP $L^{-1}$, while at ENA few samples showed freezing activity at -10 °C. This indicates the presence of a greater quantity of biogenic INPs at SGP, especially INPs active at -10 °C. It is also possible that due to the rarity of high-temperature INPs and the lower aerosol load at ENA, these INPs were present at levels below the detection limit of PINE or a WT-CRAFT system.

The median heat sensitivity of ice-active aerosol particles is shown in Figure 7. Both ENA and SGP had measurable ice nucleation above -15 °C in unheated samples, with all SGP samples having an initial freezing point above -10 °C. Once the samples were heated, no samples from ENA and only three samples at SGP showed nucleation at -10 °C. This decrease in $n_{INP}(T)$ is due to the degradation of heat-sensitive INPs in both samples, which are often referred to be of biogenic origin due to the heat sensitivity of most proteins (Hill et al., 2016; Daily et al., 2022). Only five of the 18 total filter samples from ENA showed any activation above -15 °C in the unheated samples, and none of the samples showed any activity above -15 °C once heated. Thus, the ENA samples exhibited a strong decrease in $n_{INP}$ above -15 °C once heated while Figure 7 omits to display the 100% loss. The samples from SGP showed a much greater decrease in total $n_{INP}$ at all temperatures above -15 °C, again indicating a high biogenic aerosol particle concentration in the INPs from SGP. It should be noted that samples from SGP generally had about an order of magnitude higher total $n_{INP}$ than those measured at ENA, so some of the difference between the two measurements in Figure 7 is due to this inherent difference. However, even with SGP having a higher overall concentration, the decrease in INPs after heating the SGP samples is still larger than the change seen in samples from ENA. These heat-sensitive INPs may have come from nearby SGP, as the sampling site was surrounded by agricultural land and thus fertile soil dust, or may have been transported from further away. It is presumably that the majority of INPs at SGP were terrestrial in origin based on the back trajectory analysis (see Section 3.6). Further chemical



analysis of both the bulk aerosol particle population and individual particles is needed to confirm the nature of the INPs at the two stations.

### 3.4. Comparing Online and Offline INP Measurements

When the concentration of INPs measured with PINE-3 is compared with the concentration of INPs measured with offline techniques at the same temperature it becomes apparent that the first is 1-2 orders of magnitude higher for the ExINP-ENA campaign, far outside of the range of estimated uncertainties. It is noteworthy that the sampling flow generally decreased over time, and the relative sampling flow deviation measured at the beginning and end of each sampling was on average (± standard error) larger at ENA ($27.9 \pm 3.0\%$) than that at SGP ($5.1 \pm 2.3\%$), which might hint to a decreased sampling efficiency at ENA towards the end of the sampling period. The larger deviation observed at ENA than at SGP is presumably due to longer sampling intervals (typically $\approx$ 3 days; see Table S5) as compared to SGP ($\approx$ 2 days; see Table S6), but the overall impact on $n_{INP}(T)$ deviation is not yet known. Regardless, a good agreement is observed at SGP. This is demonstrated in Figure 8. Table S7 also provides the median number concentration measured with each technique, for example, the SGP filters sampled $2.33 \pm 0.50$ L$^{-1}$ INPs active at -20 °C (measured with INSEKT), and PINE-3 measured a median $n_{INP}$ of $1.99 \pm 0.38$ L$^{-1}$. In contrast, WT-CRAFT measured the median $n_{INP}$ of 0.02 INP L$^{-1}$ at ENA for a temperature of -20 °C, while PINE-3 measured $0.36 \pm 0.05$ L$^{-1}$. This observed discrepancy between PINE-3 and WT-CRAFT data is likely due to a property inherent to the aerosol particle population at ENA, although elucidation of this property is difficult with current data. Rinaldi et al. (2021) and DeMott et al. (2018) also saw discrepancies between online and offline measurements and gave several reasons for this. Given that our samples were collected through similar inlets and onto the same filter sampling substrate, and yet samples from SGP generally did not show a mismatch between PINE and filter-based samples, the reason for the discrepancy is likely to be due to an inherent aerosol property or the environmental conditions rather than due to the measurement method. The filter samples were collected concurrently with the PINE-3 measurements, but the on-line PINE-3 measurements do not involve storing the sample for any length of time as the WT-CRAFT and INSEKT measurements do. However, there is no correlation between the length of storage time of ENA samples and $n_{INP}$ measured in the samples. There was a generally higher concentration of INPs measured with INSEKT than with WT-CRAFT, but a previous study compared WT-CRAFT





and INSEKT and concluded that their results are comparable when the same sample is examined with each method (Hiranuma et al., 2021).

The same PINE-3 instrument was used for measurements at ENA and SGP. This suggests that there is a difference in the ability of WT-CRAFT and PINE-3 when detecting the ice

nucleation ability of aerosol particles with certain chemical compositions. One possibility is that, although storage time in a -20 °C freezer might not have impacted $n_{INP}$ measured from filters substantially (Beall et al., 2020), any degradation of ice-active material occurred uniformly across the filter samples before the filters reached the storage site several thousand kilometers from the sampling site at ENA. Another possibility is that PINE-3 is known to be capable of detecting

freezing modes other than immersion freezing, while the WT-CRAFT method is only sensitive to INPs active during immersion-mode freezing processes. Möhler et al. (2021) reported that PINE-3 is capable of detecting pore condensation freezing and deposition freezing processes. It is possible that the much larger discrepancy between online and offline measurements at ENA when compared to the negligible discrepancy at SGP, is due to PINE-3 detecting additional freezing

modes that are more prominent at ENA than SGP. However, as all previous work at both sites has focused on immersion-mode freezing processes, this remains an area of uncertainty that could be examined by future researchers.

A difference in overall aerosol composition could also explain the apparent mismatch between the two sites. The samples from ENA contain much larger amounts of chloride and, due to the dominance of marine material, could be expected to contain more salts overall. While these

salt-containing particles could act as INPs if they also contained ice-active material (as salt alone is a poor immersion-mode INP), both of the offline methods employed in this study involve suspending aerosol particles in clean water. This suspension process is not present in PINE but would necessarily dissolve any soluble material that could potentially contain ice-active sites on the surface of the particles. Removal of ice-active sites following dissolution could explain the

lower $n_{INP}(T)$ seen in filter samples from ENA but not SGP, which has a lower contribution from marine sources.

### 3.5. Cloud Condensation Nuclei

Figure 9 shows the 6-hour average time series of $n_{CCN}(\%SS)$ and CCN activated fraction ($CAF = n_{CCN}(\%SS)/n_{aer}$) for ENA and SGP during our study period. While the median $n_{CCN}$ over %SS 0.1



to 0.2 is at least three times higher at SGP than ENA (Panels A and B, as well as Table S7), the overall median *CAF*s are similar at ENA (≈ 0.08 – 0.16) and SGP (≈ 0.05 – 0.15) (Panels C and D). This *CAF* similarity, as opposed to *IAF* deviation, is interesting since it suggests that the total

aerosol population both at ENA and SGP possess similar CCN activation efficiency at the measured %SS range regardless of different air mass sources, which is discussed in Section 3.6.

When $n_{CCN}$(%SS) values are compared with other meteorological data during this study, including temperature, rainfall, and windspeed, there are no apparent relationships between meteorology and $n_{CCN}$(%SS) at either ENA or SGP (tables of correlation coefficients are available

in SI S14). Although cold air outbreaks were not classified specifically, this lack of relationship with other meteorological data contrasts with the results reported by Wood et al. (2017), who found that low $n_{CCN}$(%SS) at ENA was associated with cold air outbreaks and low surface winds. It is also known that precipitation can remove aerosols through wet scavenging processes (Wang et al., 2014). The lack of a relationship between local precipitation rates at both ENA and SGP and

particle removal, particularly CCN removal, suggests that $n_{aer}$ and $n_{CCN}$(%SS) are primarily driven by transport processes rather than local aerosol generation or new particle formation. Entrainment of particles into the marine boundary layer from the free troposphere is known to occur at ENA and could be the source of some of the CCN and/or total aerosol populations (Wood et al., 2017).

Concentrations of CCN at SGP were somewhat (up to an order of magnitude) higher than

at ENA at the same supersaturations. However, as total particle concentrations were also higher at SGP, it is likely that any apparent increase was due to higher overall $n_{aer}$ rather than an overall increase in $n_{CCN}$(%SS). It is also possible that differences in the size distribution of particles could explain the difference in concentrations. However, no estimate of size distribution is available during the study period at ENA. The gap in $n_{CCN}$(%SS) data from ENA was due to instrument

malfunction, but 27 days of data are available.

At SGP, the $n_{CCN}$(%SS) data showed a similar pattern across the entire sample period, where the concentration held steady for several days and then dropped abruptly, followed by a several-day-long period of increasing concentrations until the concentration reached approximately $10^3$ cm$^{-3}$. There were variations in concentration by up to an order of magnitude at

ENA as well, but a similar pattern to that seen at SGP was not observed in the data taken at ENA. Instead, the concentration at lower supersaturations (0.1, 0.2 %SS) decreased over the course of autumn ($\rho$ = -0.64 to -0.65, p < 0.05; Table S8). This decreasing trend is notable in Figure 9A, as



well as Figure 9C. As mentioned in Section 3.2, this decrease can be attributed to a decrease in biological activity and/or total aerosol particles. Although there appeared to be patterns in variation
in the data taken at SGP, no rate of increase or decrease over the season existed.

### 3.6. Marine vs Terrestrial Aerosol Sources

HYSPLIT back trajectories origins shown in Table 1 suggest that there is little difference in the air mass origins at either inlet height, planetary boundary layer height, or cloud base height. Both
ENA and SGP showed at least 30% of trajectories originating from locations in North America south of 66 °N. While the percentage of trajectories from North America was highest for all heights from SGP, ENA showed more variation in air mass origin. The air mass back trajectory can also be used to give a semi-quantitative maximum age of the aerosols based on the occurrence of wet deposition via precipitation along the back trajectory pathway. The back trajectories reported here
were calculated for 10 days (240 hours) prior, but wet deposition was considered as well. It should be noted that if no wet deposition occurred along the trajectory, the air mass age may be greater than 240 hours. However, for the purpose of this study, it was assumed that deposition processes limited the presence of aerosols older than 240 hours (Gong et al., 2020b).

Past studies have used the HYSPLIT model to generate back trajectories from SGP (Knopf
et al., 2021; Parworth et al., 2015; Liu et al., 2021), but these studies were considering time scales of less than 10 days, rather than the 10 days used in this study. For this reason, nearly all of the identified back trajectory origins in previous work were within not just North America but within the few states surrounding the sample site. Longer back trajectories show the possibility of transported arctic and marine aerosols to the SGP site from as far away as northern Russia and the
Pacific Ocean.

The approximate maximum age of the air mass can also be determined from the rainfall amount along the backward trajectory (Gong et al., 2020b). For this study, the air mass age was assumed to be either 10 days prior to the starting time or the one-hour period at which the rainfall exceeded 7 mm. Using this method, the average air mass age at ENA (at inlet height) was $160.8 \pm$
$5.5$ hours, while the average air mass age at SGP (also at inlet height) was a slightly shorter 148.4 $\pm 5.5$ hours. There is not a large difference between the air mass ages at the two sites, but it does indicate an appreciable amount of rainfall along the air mass backward trajectories at both sites and indicates the importance of considering wet deposition when determining air mass origins, as



both sites would have much different and much longer air mass trajectories if wet deposition were not considered.

At ENA, back trajectories at all three heights show and confirm the clear marine influence indicated by the high levels of chloride, with at least 37% up to 49% of trajectories originating from the Atlantic Ocean. The next most common source was North America, ranging from 21% (at inlet height) to 35% at planetary boundary layer height. In contrast, air masses from SGP originated in the Atlantic only a maximum of 7% of the time. The Pacific Ocean was also a source of air masses at SGP, with up to 32% (cloud base height) of air masses originating there. However, it can be assumed that air masses originating in the Pacific Ocean spent time over North America to reach SGP, so they cannot be considered solely marine, unlike Atlantic Ocean air masses at ENA, which spent their entire lifetime over marine conditions.

### 3.7. Aerosol Particle Composition

Chemical composition data (not shown but the correlation analysis is discussed in SI S14) is available for the last 15 days of the autumn period at ENA and the entire sampling period at SGP. The ACSM data has been classified into four categories that include sulfate, chloride, nitrate, and organics. Due to the method used, information is only available on non-refractory chemical species. Black carbon data was also calculated and can be compared to the other chemical species present. Black carbon can be indicative of anthropogenic influence, and periods corresponding to spikes in black carbon above 50 ng/m$^3$ were removed from the overall data set to remove local anthropogenic influence (Sanchez et al., 2021). It is known that there is local anthropogenic influence at the ENA site due to its location within half a kilometer of the local airport as well as a road passing next to the site. However, there were no periods longer than three hours with such high concentrations of black carbon at ENA, indicating a predominance of clean conditions with little to no direct anthropogenic influence. Additionally, the average chloride concentration at ENA was much higher than the concentration at SGP ($0.14 \pm 0.01$ μg m$^{-3}$ and $0.03 \pm 0.002$ μg m$^{-3}$, respectively). Although chloride may come from anthropogenic sources including HCl and other reactive species, in a site located far from major anthropogenic chloride sources and directly next to the ocean, it would be expected that the chloride source is predominantly marine (Ovadnevaite et al., 2012). In contrast, most of the chloride at SGP could be expected to come from transported anthropogenic material, and the lack of marine-sourced salts provides an explanation for the much



lower overall chloride concentration (Jimenez et al., 2009). The average organic concentration at SGP was $1.16 \pm 0.06$ µg m$^{-3}$, while it was $0.75 \pm 0.14$ µg m$^{-3}$ at ENA. This higher organic concentration could explain the overall higher $n_{INP}(T)$ at SGP than ENA, although as previously discussed $n_{aer}$ could also influence the differences.

There may have been minor contributions of marine aerosols at SGP (as indicated by low

but non-zero chloride concentrations measured with the ACSM) but the non-refractory organics present at SGP are more likely to be continental than marine, due to the long distance traveled by any marine aerosols that reach the site and the potential for mixing with transported continental organics during this transportation process. The chloride concentration at ENA was more than an order of magnitude higher, indicating a much stronger marine signal, while the organic

concentration was only slightly lower. To reach the sampling site air masses must spend several days or longer over continental regions, including passing over the Great Plains region, a known dust source. For this reason, dust and local soil dust are considered the major aerosol types reaching SGP based on both our back trajectories and previous studies (Knopf et al., 2021; DeMott et al., 2015).


## 4    Discussion

It would be expected that the sources of CCN would vary between terrestrial sites and those dominated by marine aerosols. A possible CCN source, especially at ENA, is aerosols formed via new particle formation (NPF), either locally or formed elsewhere and entrained downward toward

ground level over the measurement area. NPF has been described at ENA by Zheng et al. (2018), who found that the growth of free troposphere particles in the marine boundary layer was a source of CCN at ENA. Although this was found to be especially true in the summer, this process was also a source of autumn CCN. NPF has also been described by several previous studies at SGP (Hodshire et al., 2016; Chen et al., 2018; Marinescu et al., 2019). However, the NPF was not

attributed to the same entrainment processes seen at ENA. Instead, NPF at SGP occurs via a wide variety of pathways including the growth of biogenic highly oxygenated materials. These small particles grown via NPF can then activate as CCN. However, our methods cannot differentiate between CCN formed via NPF and those small particles released via primary aerosol production. Additionally, transported secondary aerosols can also be within the same size range as those

formed via NPF.





The most striking pattern between aerosol properties occurs between CCN at lower supersaturations of 0.1 and 0.2% SS and $n_{INP}(T)$ measured with PINE-3. This data is summarized in Table 2 and also shown in Figure 10 for ENA and SGP. When considering Spearman correlation coefficients, all p-values at ENA are less than 0.05, while at SGP all are greater than 0.05.

Additionally, all $\rho$-values at SGP are less than 0.5, which also indicates a lack of relationship between CCN and INPs at SGP. There is a positive relationship between CCN and INP at ENA at all INP nucleation temperatures and CCN supersaturations, with the strongest relationship at -20 °C and 0.1 %SS. This indicates that as $n_{CCN}(\%SS)$ increases, so does $n_{INP}(T)$ at ENA, but that there is no such relationship seen at SGP and in no other study before, to our knowledge. There is also

a significant linear relationship between CCN and INPs active at -20 °C at ENA, as the Pearson correlation coefficient is 0.66 and 0.59 for 0.1% and 0.2% supersaturation, respectively. However, it would not be expected that the same particles are first activating as CCN with the condensation of supercooled water and then acting as INPs by catalyzing freezing, a freezing process known as condensation freezing. Past studies have estimated that only a very small percentage of all

atmospheric ice nucleation events undergo condensation-freezing processes (Westbrook and Illingworth, 2011). CCN can also activate even following dissolution, while INPs must maintain their ice-active structure and surface characteristics to maintain their ice nucleation ability. The strong correlation at ENA does not definitively prove that the INP and CCN populations are the same particles, but the abundance of both sea salts (a known CCN source) and organics (known

ice-active material) at ENA as seen in bulk chemistry suggests that the ice-active and CCN-active material may be internally mixed within the particles present at this location.

This relationship between CCN and INPs is unique to ENA and has not been observed at other sites where aerosols would be considered to have marine-dominant sources, including other island sources (Gong et al., 2020b). Although comparative aerosol particle chemistry was not

examined in detail during this study, future work at ENA should consider both the composition and concentration of aerosol particles, especially those that are ice and or cloud active. Previous studies have examined the physicochemical properties of aerosol particles at ENA with either long-term multi-year data from an ACSM or shorter-term data from an aerosol mass spectrometer (AMS) (Zawadowicz et al., 2021). Both of these instruments have also been deployed at SGP (Liu

et al., 2021; Subba et al., 2021), including at times when CCN and INPs have been studied (Knopf et al., 2021). However, there is a distinct lack at both sites of long-term, high time resolution data



examining INPs, CCN, and particle chemistry concurrently, so this should be a priority for future campaigns in order to determine the precise physicochemical properties of ENA aerosols that enable the same population to act as CCN and INPs.


## 5 Conclusions

This study is the first high-time resolution comparison of aerosol particles, CCN, and INP from ENA and SGP, using the two sites as contrasting terrestrial and marine sites. The overall measured $n_{INP}(T)$ at SGP was approximately three times higher than the concentration measured at ENA

based on an online INP monitor. Although $n_{aer}$ at SGP was also larger than $n_{aer}$ at ENA, the increased concentration at SGP alone did not explain the higher $n_{INP}(T)$ at SGP. An unprecedented relationship between $n_{INP}(T)$ and $n_{CCN}(\%SS)$ was seen in ENA at lower supersaturations that are more representative of natural systems. This relationship, found based on the highly time-resolved $n_{INP}(T)$ measurements, suggests the presence of a population of aerosols containing both salts and

organic material capable of acting as CCN and organic material and mineral dust capable of acting as INPs, but more information about the exact composition of CCN and INPs at the sites is needed. Future studies could focus on both bulk measurements over long time scales with instruments such as the ACSM and bulk aerosol mass spectrometry, and single particle chemistry to investigate particle mixing-state via methods including scanning electron microscopy- x-ray dispersion (SEM-

EDX), single particle mass spectrometers, and scanning transmission x-ray microscopy coupled with near-edge x-ray absorption fine structure (STXM-NEXAFS).

While we see this relationship at the ENA site, the SGP site does not show any relationship between CCN and INP, suggesting that the marine-dominant nature of the aerosols at ENA is driving this relationship. However, further information on the drivers of this relationship should

be found via controlled laboratory studies.

In addition to having observed fewer INPs at ENA, the sampled INPs have shown to not be sensitive to heating, while the INPs sampled at SGP are. This heat sensitivity of particles at SGP is especially apparent at higher nucleation temperatures. Heat sensitivity is often attributed to organic and/or biogenic material, which can be from soil dust at SGP. In contrast, samples from

ENA, while still containing organic material, may contain a higher proportion (but not a higher number concentration) of mineral dust particles, which can have a composition that does not degrade or lose ice nucleation ability with heat.



Additional similar sites both in the mid-latitude Atlantic Ocean and other mid-latitude oceans should be identified for further studies, with a particular focus on high-time-resolution INP measurements (8 to 12 minutes) coupled with CCN measurements to determine if this relationship is unique to the ENA site or is replicated at other marine boundary layer sites. This coupling may represent an unexplored relationship that should be parameterized and added to global climate models to better represent aerosol populations within natural systems and hopefully reduce error in predicted radiative effects of global aerosols. It is apparent that organic material (as seen in samples from SGP) is capable of acting as INPs, but the type of INPs at ENA must be better understood by increasing both spatiotemporal sampling resolution and physicochemical sampling of individual INPs. These broad goals open up the possibility for and demonstrate the need for partnership and collaboration between different research groups to understand this complex problem.

**Acknowledgments:**

This research was supported by the US Department of Energy, Office of Science, Office of Biological and Environmental Research (grant no. DE-SC-0018979). The authors gratefully acknowledge the NOAA Air Resources Laboratory (ARL) for the provision of the HYSPLIT transport and dispersion model and/or READY website (https://www.ready.noaa.gov) used in this publication. The authors acknowledge Tercio Silva, Bruno Cunha, Carlos Sousa, Pawel Lech, Hannah Frances Ransom, John Archuleta, Heath Powers, and Karen Caporaletti for their onsite and administrative contributions to the ExINP-ENA campaign. Our half a year campaign at the ENA site could not be completed without their support. The authors also acknowledge the contributions of the SGP onsite technical team (Chris Martin, Mark Smith, and Ken Teske) and administrative team (John Schatz, James Martin, George Sawyer, David Swank, Tim Grove, Rod Soper, Judy Brooke, and Michael T. Ritsche). Naruki Hiranuma and Elise K. Wilbourn thank Kimberly Sauceda for her contribution to the inlet loss measurement. The authors appreciate Romy Fösig and Nicole Büttner for the useful discussion regarding the PINE data processing.

**Data availability:**

Original data created for the study will be available in the Supplement and at https://www.doi.org/10.6084/m9.figshare.24199176 (Hiranuma, 2023).



**Supplement:**

795 The supplement related to this article is available online at: www.atmospheric-chemistry-and-physics.net

**Authors Contribution:**

NH designed the concept of this collaborative research. EKW led the writing of the manuscript

800 with the support of all authors. The methodology was developed by OM, LL, KH, JN, and NH. The onsite and remote measurements at SGP were conducted by HSKV and NH. The ENA experiment was remotely conducted by EW and NH, followed by the initial onsite setup led by LL. Offline INP measurements were supervised by KH and NH and contributed by CG, HSKV, and EW. Formal data analyses were led by EW.


**Competing Interests:**

The authors declare no conflict of interest.





TABLES


**Table 1: Percentage of air masses originating from each location, determined from 240-hour HYSPLIT back trajectories (back trajectories may be younger than 240 hours if rainfall exceeded 7mm). Back trajectories were calculated at three different heights for each 6-hour sample period. The dominant air mass origin at each height for each location is in bold.**

| ORIGIN | ENA (2020) | | | SGP (2019) | | |
|---|---|---|---|---|---|---|
| | Inlet Height | Planetary Boundary Layer Height | Cloud Base Height | Inlet Height | Planetary Boundary Layer Height | Cloud Base Height |
| Arctic Circle | 4.6 | 3 | 0.8 | 9.6 | 3.7 | 3.2 |
| Arctic Ocean | 7.6 | 4.3 | 0 | 6.9 | 9.1 | 1.1 |
| Atlantic Ocean | **49.4** | **38.3** | **37.3** | 6.9 | 3.7 | 3.2 |
| Europe | 3.8 | 2.6 | 3 | 0 | 0 | 0 |
| Greenland & Iceland | 2.5 | 1.7 | 2.1 | 0 | 0 | 0 |
| Gulf of Mexico | 0 | 0 | 0 | 2.7 | 2.1 | 5.4 |
| Latin America | 0 | 0 | 0 | 0 | 0 | 5.4 |
| Marginal Arctic Ocean | 9.7 | 6.4 | 3.8 | 2.7 | 3.2 | 0 |
| North America | 20.7 | 35.3 | 31 | **60.1** | **51.9** | **48.4** |
| Norwegian Sea | 0.8 | 2.1 | 1.7 | 0 | 0 | 0 |
| Pacific Ocean | 0.4 | 3.4 | 9.7 | 12.8 | 22.5 | 32.3 |
| Russia | 0.4 | 2.6 | 5.5 | 1.1 | 3.7 | 1.1 |
| Western Africa | 0 | 0.4 | 2.5 | 0 | 0 | 0 |




**Table 2: The correlation coefficients calculated between $n_{CCN}$(%SS) and $n_{INP}(T)$ at (A) ENA and (B) SGP (Spearman rank-sum correlations, $\rho$, and Pearson coefficient, $r$). Both data sets are time-averaged for 6 hours. All data of $\rho$ >0.3 has a p-value <0.05 and is considered statistically significant.**


| **A. ENA** | | | | |
|---|---|---|---|---|
| | Spearman | | Pearson | |
| Freezing Temperature (°C) | 0.1 %SS | 0.2 %SS | 0.1 %SS | 0.2 %SS |
| -30 | 0.41 | 0.41 | -0.14 | -0.06 |
| -25 | 0.77 | 0.76 | 0.17 | 0.14 |
| -20 | 0.81 | 0.80 | 0.66 | 0.59 |

| **B. SGP** | | | | |
|---|---|---|---|---|
| | Spearman | | Pearson | |
| Freezing Temperature (°C) | 0.1 %SS | 0.2 %SS | 0.1 %SS | 0.2 %SS |
| -30 | -0.01 | 0.1 | -0.03 | 0.12 |
| -25 | 0.09 | 0.03 | 0.05 | 0.004 |
| -20 | -0.21 | 0.04 | 0.001 | 0.06 |




FIGURES


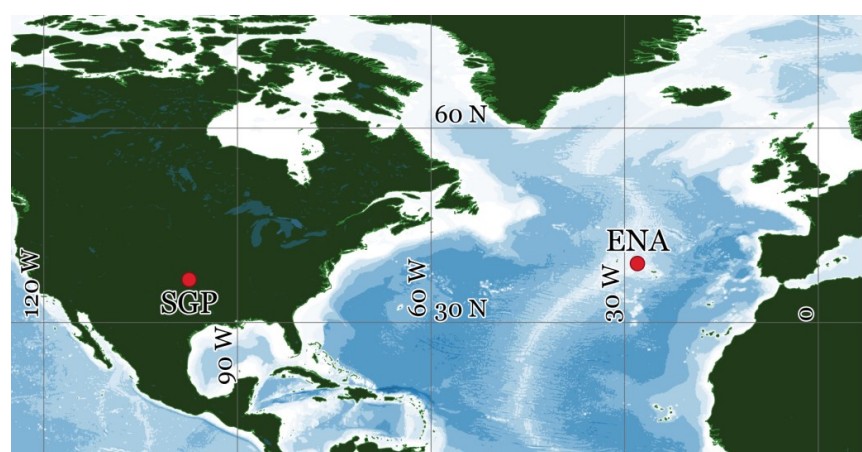

**Figure 1: The two DOE ARM site locations: the ENA site is located on Graciosa Island, Azores, Portugal, and the SGP site is located in Oklahoma in the United States.**






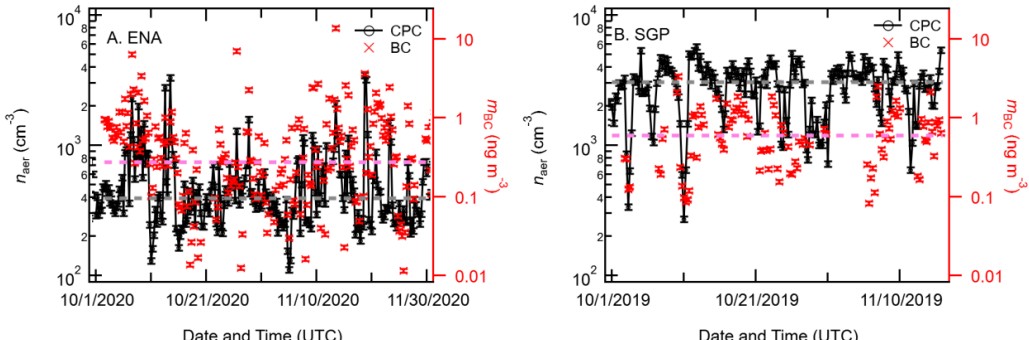

**Figure 2: The average total particle concentration ($n_{aer}$, cm$^{-3}$, shown with black dots), measured with a CPC and plotted at 6-hour averaged intervals. The red crosses indicate the 6-hour average mass of black carbon**
**($m_{BC}$, ng/m$^3$). Data for the ENA study period (2020) is plotted in panel A, and data for the SGP study period (2019) is plotted in panel B. In both panels, $n_{aer}$ is plotted in black on the left-hand y-axis while the black carbon mass is plotted in red on the right-hand y-axis. Dashed lines represent median values for the measured periods. The measurement accuracy of ARM-CPC and BC concentrations is presumably dominated by inlet flow variability of 5% (Kuang, 2016).**




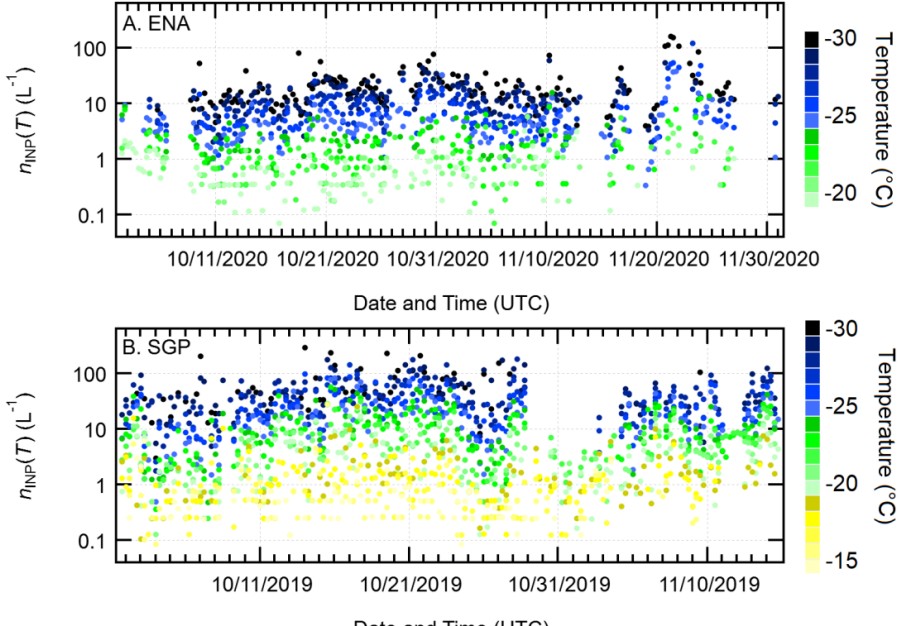

**Figure 3: INP concentrations ($n_{INP}(T)$) measured at the ENA and SGP sites with the PINE-3 system. Panel A shows data from the ENA sampling site (collected in 2020), while panel B shows data from SGP (collected in 2019). Each point represents a 6-hour time-averaged concentration, while the color scale indicates the freezing temperature.**





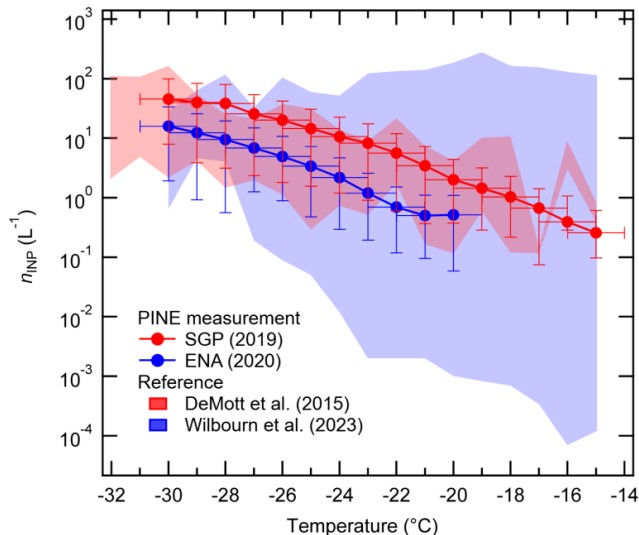

**Figure 4: Median INP concentrations at each °C at ENA (2020, blue dots) and SGP (2019, red dots). The horizontal error bar represents the temperature uncertainty from PINE-3 and the vertical error bars represent the standard error. The red-shaded area shows the maximum and minimum $n_{INP}(T)$ measured by DeMott et al. (2015), while the blue shape represents the maximum and minimum $n_{INP}(T)$ measured by all previous studies of INPs from marine dominant sites located in the Atlantic Ocean (see Wilbourn et al. (2023) for more information).**





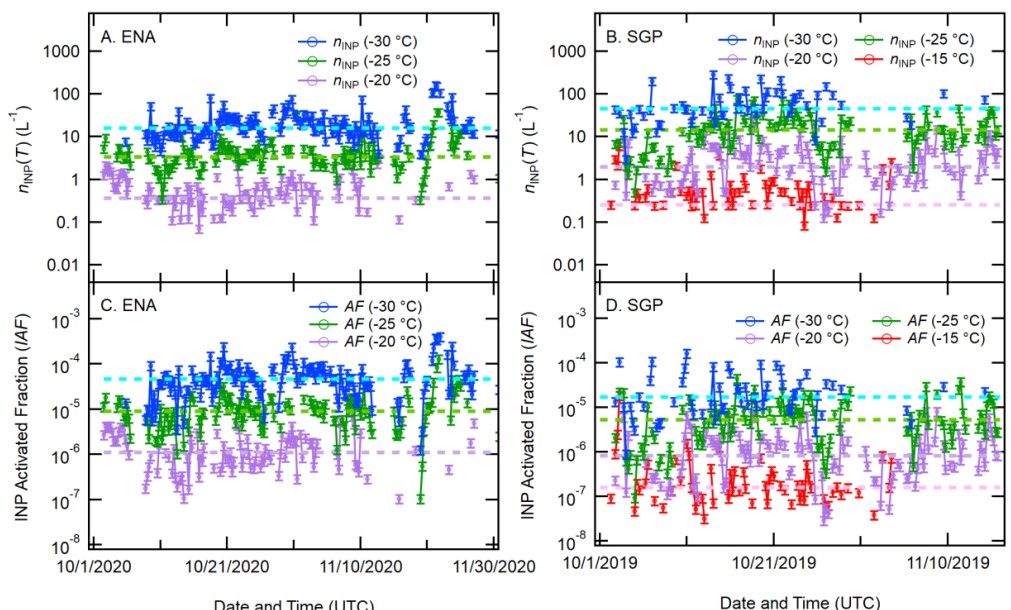

**Figure 5: The 6-hour time-averaged $n_{INP}(T)$ at selected temperatures at ENA (A) and SGP (B). Panels C and D**
**display corresponding $IAF$ (= $n_{INP}(T)/n_{aer}$) at ENA and SGP, respectively. Dashed lines represent median values**
**for the measured periods. Error bars in $n_{INP}$ are represented by the systematic error (± 20%). Errors in $IAF$**
**are estimated as ± 21% (= $\sqrt{20^2 + 5^2}$). Note: The 5% error stems from Kuang (2016).**


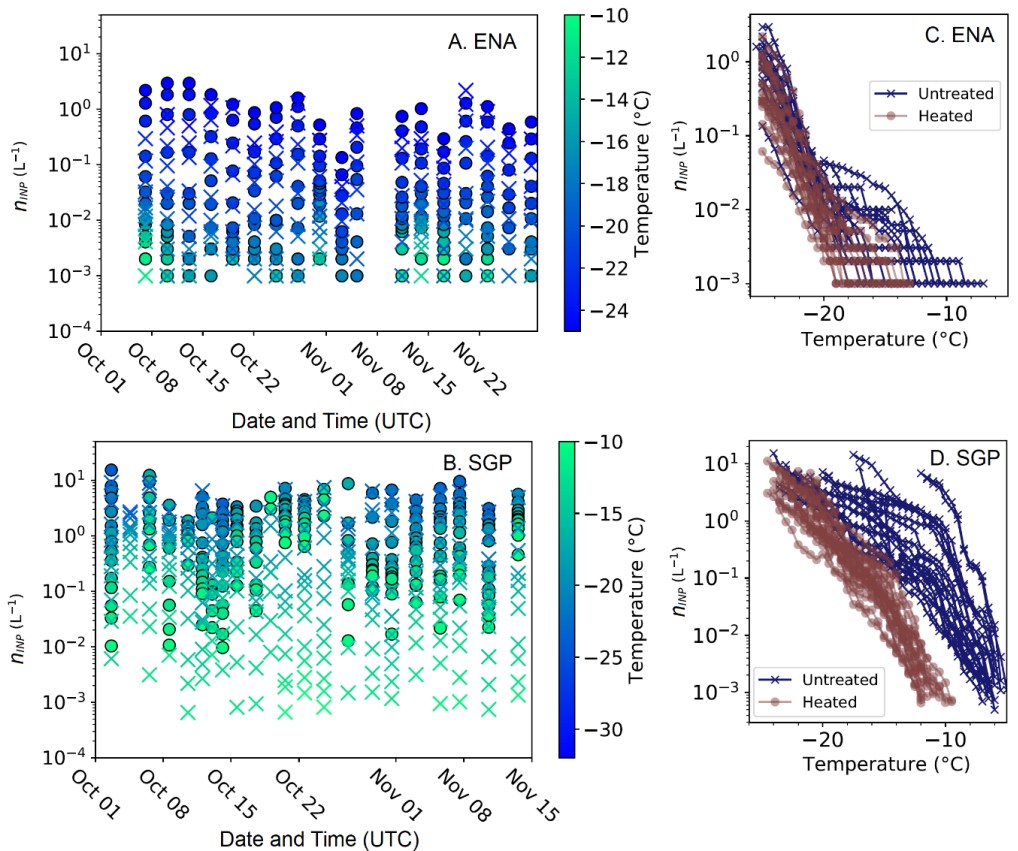

**Figure 6: Ice nucleating particle concentrations ($n_{INP}$, plotted with circles) measured from samples collected onto 47 mm polycarbonate filters over approximately 4-day time periods, along with $n_{INP}$ concentrations measured after heating the samples to 100 °C for 20 minutes (plotted with 'X's). The data for each filter is plotted at the midpoint of the sampling period, and the color of each point represents the freezing temperature. Data from the ENA site (2020) is plotted in panel A, and data from the SGP site (2019) is plotted in panel B. A summary of freezing spectra of both untreated and heated samples from ENA and SGP is shown in panels C and D, respectively.**



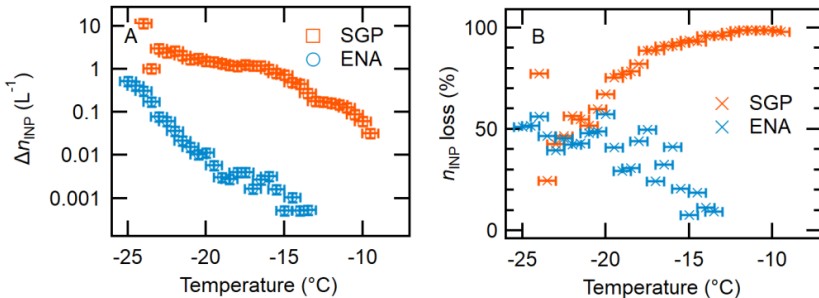

**Figure 7: Median change in $n_{INP}(T)$ after heating for the data from ENA (blue circles) and SGP (orange crosses).**
**The changes in absolute $n_{INP}(T)$ and percentage for $\Delta T$ of 0.5 °C are shown in panels A and B, respectively. The uncertainties are ± 0.5 °C with ± 23.5% for temperature and $n_{INP}$. Note: The 100% $n_{INP}$ loss is not displayed in Panel B.**




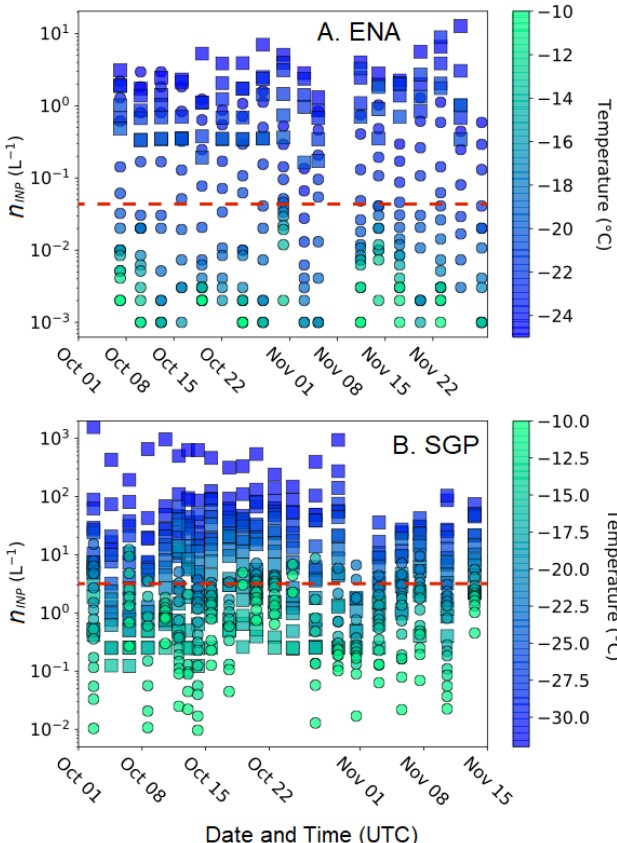

**Figure 8: Ice nucleating particle concentrations ($n_{INP}$. std. L$^{-1}$ air) measured from 4-day samples collected on filters (plotted with dots) and average concentration of INPs measured with PINE-3 over the same time period each filter sample was collected (plotted with squares). The color bar indicates the freezing temperature (same color bar can be used for PINE-3 and filter data) and the dashed red line indicates the median $n_{INP}$(-15 °C) as measured with PINE-3. Data from 2020 at ENA is plotted in panel A while data from 2019 at SGP is plotted in panel B.**





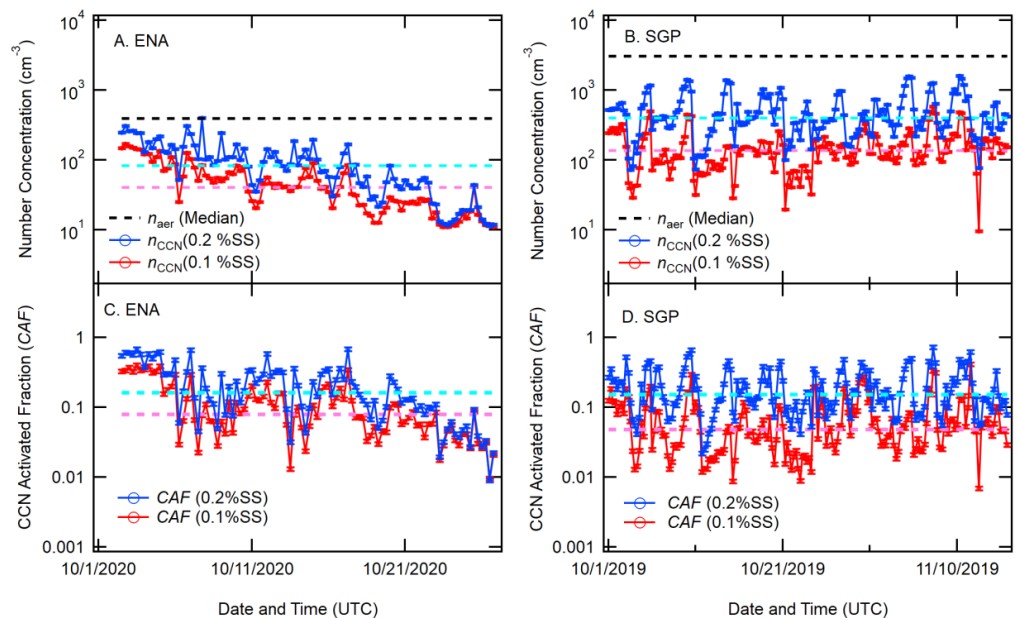

**Figure 9: The 6-hour time-averaged CCN concentrations, measured with 0.1%SS (red) and 0.2%SS (blue) at ENA (A) and SGP (B). Dashed lines represent median values for the measured periods (summarized in Table**
**S7). The reference $n_{aer}$ median line (black dashed line), adapted from Figure 2, is superposed in panels A and B to guide the eye. The $CAF$ (= $n_{CCN}$(%SS)/$n_{aer}$) time series for each %SS measured at ENA and SGP are shown in panels C and D. The measurement accuracy of CCN measurement included in this figure is ±3% (Uin, 2022). The uncertainty in $CAF$ is estimated as ± 6% (= $\sqrt{3^2 + 5^2}$). Note: The 5% error stems from Kuang (2016) for a stack inlet flow variability.**




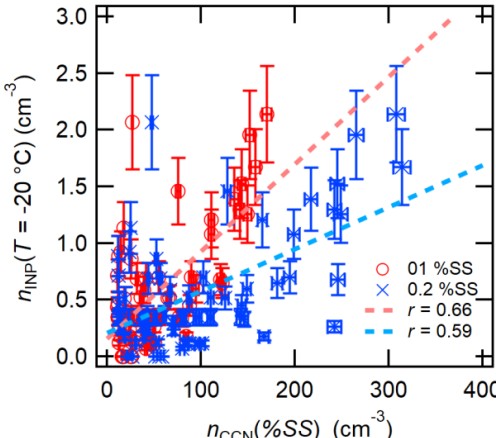

**Figure 10: The relationship between $n_{CCN}$ measured at two supersaturation (SS) conditions and $n_{INP}$ active at -20 °C measured at ENA [$n_{INP}$(-20 °C) = 0.0077 x $n_{INP}$ (0.1%SS) + 0.1591; $n_{INP}$(-20 °C) = 0.0037 x $n_{INP}$(0.2%SS) + 0.2065]. The measurement accuracy of INP and CCN measurement included in this figure is ±20% (Möhler et al., 2021) and ±3% (Uin, 2022).**



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
