# Peer review of "Measurement Report: A comparison of ground-level icenucleating particle abundance and aerosol properties during autumn at contrasting marine and terrestrial locations"

_EGUsphere, 2023_

## Referee Comment (RC2)

**General comments:**

The manuscript provides an intriguing exploration of the sources and properties of INPs and CCNs, while the study addresses an intriguing and important topic, significant improvements in clarity, structure, grammar, data analysis, and the careful drawing of conclusions are necessary to meet the rigorous standards of scientific publication.

**Major comments:**

1. The manuscript is challenging to follow. The scientific objectives are not articulated clearly and lack a defined scope. It's crucial for the manuscript to provide a clear and comprehensive explanation of the relationship between CCN and INPs. As immersion freezing INPs are initially CCN before acting as nuclei for ice crystal formation, What does a strong correlation imply? This issue is compounded by a disjointed logical structure, making it difficult for readers to follow the progression of your study. In addition, a well-organized manuscript with a clear introduction, methodology, results, and conclusion is critical needed.

2. The manuscript's conclusions are replete with conjectures and assumptions not robustly supported by the data presented. e.g. how can the author conclude the influence of the mineral dust on INPs at ENA without any aerosol size distribution and also no chemical components. Scientific studies should draw conclusions directly and cautiously from the results, avoiding overgeneralization and unwarranted speculation. It's essential to clearly state the limitations of your study and discuss the conclusions within the context of these limitations.

3. The analysis presented appears to be superficial and does not delve deeply into the complexities of the data. A more rigorous and detailed statistical analysis is crucial to understand the nuances and implications of your findings fully.

**Specific comments:**
1. In the Introduction, mischaracterize immersion freezing mechanisms INPs. The immersion freezing should also include the condensation freezing.
2. Regarding the comparison between PINE and offline freezing droplet measurements, it appears that no direct comparison has been provided. From the current presentation in Figure 6 and Figure 8, at least, it is not discernible. These figures should be redrawn to clearly illustrate the comparison and provide a more direct and insightful analysis of the results.
3. About BC measurement using PASP, 'with the absorption at 529 nm most 330 representative of black carbon aerosols'. The PASP can provide aerosol absorption coefficients at three wavelength, the red light can represent BC characteristic more.

---

## Author Response (AR1)

RC1

**Comment on "Measurement Report: A comparison of ice-nucleating particle and cloud condensation nuclei sources and properties during autumn at contrasting marine and terrestrial locations" by Wilbourn et al.**

This manuscript presents aerosol, cloud condensation nuclei, and ice-nucleating particle measurements at two contrast sites, i.e., the marine site at ENA and the continental site at SGP. The description of the measurement is clear and comprehensive. There are concerns regarding the data cleaning and clarity of data result interpretation. Addressing the following major comments is imperative before the manuscript can be considered for publication:

AR: The authors appreciate the comments. We believe that the readability and the quality of this paper have improved with the changes made to the current version of the manuscript. Below, we provide our point-by-point responses.

Major comments

1. Data cleaning is missing in this study. Do you consider the CPC data cleaning at the ENA site? In line 382, the total particle number concentration should be much lower than ~3000 cm$^{-3}$ after data cleaning. A previous study by Gallo et al 2020 (https://acp.copernicus.org/articles/20/7553/2020/) has shown that the ENA site is very often polluted.

AR: L353-356 – The authors thank the referee for providing us with a useful reference. Concerning the anthropogenic influence, especially at ENA, we considered the CPC data cleaning by means of black carbon mass concentrations ($m_{BC}$) as a pollution indicator and excluded notable $m_{BC}$ spikes. We now clarified this point with the suggested reference in Sect. 2.5.2 – "Black carbon can be indicative of anthropogenic influence. For instance, at ENA, due to airport operations, a minute average $n_{aer}$ can instantaneously exceed 8000 cm$^{-3}$ (Gallo et al., 2020). Therefore, the periods corresponding to spikes in black carbon above 50 ng/m$^3$ were removed from the overall data set to remove local anthropogenic influence (Sanchez et al., 2021)." With this data cleaning process, we were able to remove notably high $n_{aer}$ reported in Gallo et al. (2020). As a result, our median $n_{aer}$ is 393.25 cm$^{-3}$ as stated in Sect. 3.1. This number is within the seasonal baseline $n_{aer}$ values of 346 cm$^{-3}$ (winter) to 428 cm$^{-3}$ (summer) from the Aerosol and Cloud Experiments campaign in 2017 as reported by Gallo et al. (2020).

In addition, in Sect. 1 (L399-404), we added the following sentences – "This number at ENA is within the seasonal baseline $n_{aer}$ values of 346 cm$^{-3}$ (winter) to 428 cm$^{-3}$ (summer) from the ACE-ENA campaign in 2017 (Gallo et al., 2020). Even long-term ENA-CPC data from 2015 to 2021 support the seasonal variation between $\approx$ 300 cm$^{-3}$ (winter) and $\approx$ 600 cm$^{-3}$ (summer) at ENA (Ghate et al., 2023). At SGP, our median $n_{aer}$ of 3055 cm$^{-3}$ is similar to the previous total aerosol abundance measured at SGP for air masses flow from typical Midwest (2304 cm$^{-3}$) and Northwest (3369 cm$^{-3}$) in May 2003 reported in Wang et al. (2006)."

Minor comments

1. L 41: -32 ºC or -38 ºC for the homogeneous freezing?

AR: L60-61 – Corrected. New sentence reads as " to approximately -35 °C (238 K) or below."

2. L528: It is unclear to me what agreement (i.e., agreement between what and what) you are referring to.

AR: Figure 8 & associated text in L595 onward – We now clarify a good agreement between online and offline $n_{INP}$ measurements is observed at SGP in Sect. 3.4.

[Figure]

The figure above shows the comparison of online vs. offline $n_{INP}(T)$ spectra with blue and red box plots, respectively, from SGP. The online PINE-3 spectra from SGP are time-averaged for 48-hour to match with the aerosol sampling time intervals for the offline INP analysis. The offline spectra are all non-heated data. Individual boxes display median (orange line), average (green line), 25% & 75% percentile (whiskers), and outliers beyond 5% & 95% percentile values. The grey-shaded area shows the maximum and minimum $n_{INP}(T)$ measured by DeMott et al. (2015) for SGP.

3. L554-555: My impression is that PINE-2 usually measures immersion freezing INPs, as the supersaturation created inside the chamber. Please correct me if I am wrong.

AR: L640-650 – The referee is right that PINE mainly measures immersion freezing INPs. However, PINE can measure INPs formed through pore condensation freezing and deposition freezing processes at ice supersaturation yet water subsaturated conditions. More clarification is provided in Sect. 3.4. The authors also made our clear point that the deposition mode was missing in offline analyses.

4. L575-580: The CAF values show large variations for two sites. Only comparing the median value is not reasonable. Better compare the probability density function of CAF at two sites.

AR: The authors agree. Considering we do not have size-dependent chemical composition measurements, the authors realize that the comparison of CCN and INP is complex to deduce valid conclusions in this measurement report. We removed Section 4 Discussions, SI Section S14, and Figure 10, which discussed this INP-CCN comparison. We also removed CCN discussions from the Abstract, Section 3.5, Fig. 9, Table 2, and Table S7.

L456 -: We revised the manuscript focusing on INP research. Such long-term INP data from SGP and ENA are unique and novel. The revised manuscript discusses the trends and general comparison between the two sites. We considered the probability density function of INP concentrations and ice nucleation efficiency, $n_s$, in the revised manuscript in Sect. 3.2 (L456-). We also changed the title to "Measurement Report: A

comparison of ground-level ice-nucleating particle abundance and aerosol properties during autumn at contrasting marine and terrestrial locations" to represent the revised content and edits we made according to the referee's suggestions and comments.

5. L608: Change "can be" to "is likely".

AR: Due to the reason addressed above, this part is now removed from the manuscript.

6. A recent study by Ghate et al. 2023 (https://agupubs.onlinelibrary.wiley.com/doi/pdf/10.1029/2023JD038636) should be discussed when discussing the aerosol and CCN at the ENA site.

AR: L400-402 – Thank you for this useful reference. We now added, "Even long-term ENA-CPC data from 2015 to 2021 support the seasonal variation between $\approx 300$ cm$^{-3}$ (winter) and $\approx 600$ cm$^{-3}$ (summer) at ENA (Ghate et al., 2023)." in Sect. 3.1.

**Citation**: https://doi.org/10.5194/egusphere-2023-1456-RC1

References

DeMott, P. J., Suski, K. J., Hill, T. C. J., and Levin, E. J. T.: Southern Great Plains Ice Nuclei Characterization Experiment Final Campaign Summary, DOE Office of Science Atmospheric Radiation Measurement (ARM) Program, Washington DC, USA, DOE/SC-ARM-15-012, https://www.arm.gov/publications/programdocs/doe-sc-arm-15-012.pdf, (last visited: February 27th, 2024), 2015

Gallo, F., Uin, J., Springston, S., Wang, J., Zheng, G. J., Kuang, C. A., Wood, R., Azevedo, E. B., McComiskey, A., Mei, F., Theisen, A., Kyrouac, J., and Aiken, A. C.: Identifying a regional aerosol baseline in the eastern North Atlantic using collocated measurements and a mathematical algorithm to mask high-submicron-number-concentration aerosol events, Atmospheric Chemistry and Physics, 20, 7553-7573, https://doi.org/10.5194/acp-20-7553-2020, 2020.

Ghate, V. P., Surleta, T., Magaritz-Ronen, L., Raveh-Rubin, S., Gallo, F., Carlton, A. G., and Azevedo, E. B.: Drivers of cloud condensation nuclei in the Eastern North Atlantic as observed at the ARM site, Journal of Geophysical Research: Atmospheres, 128, e2023JD038636, https://doi.org/10.1029/2023JD038636, 2023.

Sanchez, K. J., Zhang, B., Liu, H. Y., Saliba, G., Chen, C. L., Lewis, S. L., Russell, L. M., Shook, M. A., Crosbie, E. C., Ziemba, L. D., Brown, M. D., Shingler, T. J., Robinson, C. E., Wiggins, E. B., Thornhill, K. L., Winstead, E. L., Jordan, C., Quinn, P. K., Bates, T. S., Porter, J., Bell, T. G., Saltzman, E. S., Behrenfeld, M. J., and Moore, R. H.: Linking marine phytoplankton emissions, meteorological processes, and downwind particle properties with FLEXPART, Atmospheric Chemistry and Physics, 21, 831-851, https://doi.org/10.5194/acp-21-831-2021, 2021.

Wang, J., Collins, D., Covert, D., Elleman, R., Ferrare, R. A., Gasparini, R., Jonsson, H., Ogren, J., Sheridan, P., and Tsay, S. C.: Temporal variation of aerosol properties at a rural continental site and study of aerosol evolution through growth law analysis, Journal of Geophysical Research-Atmospheres, 111, D18203, https://doi.org/10.1029/2005jd006704, 2006.

RC2

**General comments:**

The manuscript provides an intriguing exploration of the sources and properties of INPs and CCNs, while the study addresses an intriguing and important topic, significant improvements in clarity, structure, grammar, data analysis, and the careful drawing of conclusions are necessary to meet the rigorous standards of scientific publication.

AR: The authors thank Referee #2 for the peer-review comments, which motivated further analyses and improved the overall presentation. We found these comments as invaluable guidance to further improve the manuscript. We believe that the analysis and discussion, as well as revised figures and tables, in the revised manuscript (including the SI materials) are robust. We have very good data from our >45 days survey from SGP and ENA. The authors admit that some insufficient discussions might have led some of our data interpretation and presentation in an original manuscript to be unclear. To allay the reviewer's concerns and mitigate any misgivings, the authors decided to change the title of the manuscript to "Measurement Report: A comparison of ground-level ice-nucleating particle abundance and aerosol properties during autumn at contrasting marine and terrestrial locations". The authors think this title represents the revised content and edits we made according to the referee's suggestions and comments. We have also revised our abstract, the Summary and Outlook section, all figures & tables, and the overall structure to reflect all of our major revisions and to increase the readability of this paper with rigorous analysis and discussion. Additional citations were used to reduce speculations in part. Typos and grammar have been checked and corrected by a native English speaker. Below, we provide our point-by-point responses.

**Major comments:**

1. The manuscript is challenging to follow. The scientific objectives are not articulated clearly and lack a defined scope.

AR: L19-: Our research objective is to analyze the correlation between sources and abundance of INPs in different environments. We now clearly discuss definite goals for this study. The revised abstract states "our study applied an *in situ* forced expansion cooling device to measure ambient INP concentrations and test its automated continuous measurements at atmospheric observatories, where complementary aerosol instruments are heavily equipped. Using collocated aerosol size, number, and composition measurements from these sites, we analyzed the correlation between sources and abundance of INPs in different environments. Toward this aim, we have measured ground-level INP concentrations at two contrasting sites, one in the Southern Great Plains (SGP) region of the United States with a substantial terrestrially influenced aerosol population, and one in the Eastern North Atlantic Ocean (ENA) with a primarily marine-influenced aerosol population. These measurements examined INPs mainly formed through immersion freezing and were performed at a ≤ 12-minute resolution and with a wide range of heterogeneous freezing temperatures (*T*s above -31 °C) for at least 45 days at each site. The associated INP data analysis was conducted in a consistent manner. We also explored the additional offline characterization of ambient aerosol particle samples from both locations in comparison to *in situ* data."

In the Introduction section (L77-), the purpose & goals of this study are also now clarified. "This study reports the automated continuous measurements of ambient INP concentrations in comparison to offline INP abundance measurements in a wide range of heterogeneous freezing temperatures from two field campaigns, including Examining the Ice-Nucleating Particles from Southern Great Plains (ExINP-SGP; https://www.arm.gov/research/campaigns/sgp2019exinpsgp) and from Eastern North Atlantic (ExINP-ENA; https://armweb0-stg.ornl.gov/research/campaigns/ena2020exinpena). The goals of this

study are to quantify INPs continuously measured for >45 days at the two ground observatories located in unique ambient conditions (i.e., predominantly terrestrial and marine-influenced sites) and to understand the properties of immersion-mode INPs with respect to the origin of air mass and ambient aerosol properties (i.e., number and surface area concentrations, as well as chemical composition)."

The expected outcome of this study is also now refined (L136-141). "The resulting INP data from both sites were processed and analyzed in a consistent manner to elucidate INP sources and abundance in different environments and set a basis for long-term INP data processing and analysis at more remote locations in the future. The high-time resolution data, time-averaged data, and temperature-binned data products here provide first-of-its-kind information about INPs that could be useful for global climate models to reduce the uncertainties associated with current aerosol measurements (Murray et al., 2021)."

It's crucial for the manuscript to provide a clear and comprehensive explanation of the relationship between CCN and INPs. As immersion freezing INPs are initially CCN before acting as nuclei for ice crystal formation, What does a strong correlation imply? This issue is compounded by a disjointed logical structure, making it difficult for readers to follow the progression of your study. In addition, a well-organized manuscript with a clear introduction, methodology, results, and conclusion is critical needed.

AR: The authors agree. Considering we do not have size-dependent chemical composition measurements, the authors realize that the comparison of CCN and INP is complex to deduce valid conclusions in this measurement report. We removed Section 4 Discussions, SI Section S14, and Figure 10, which discussed this INP-CCN comparison. We also removed CCN discussions from the Abstract, Section 3.5, Fig. 9, Table 2, and Table S7. We revised the manuscript focusing on INP research and modified all figures and tables considering the referee's comments. Such long-term INP data from SGP and ENA are unique and novel. The revised manuscript discusses the trends and general comparison between the two sites. We considered the probability density function of INP concentrations and ice nucleation efficiency, $n_s$, in the revised manuscript. The overall manuscript structure was majorly revised, and the revisions are highlighted.

2. The manuscript's conclusions are replete with conjectures and assumptions not robustly supported by the data presented. e.g. how can the author conclude the influence of the mineral dust on INPs at ENA without any aerosol size distribution and also no chemical components. Scientific studies should draw conclusions directly and cautiously from the results, avoiding overgeneralization and unwarranted speculation. It's essential to clearly state the limitations of your study and discuss the conclusions within the context of these limitations.

AR: The referee mentions the paper has multiple conjectures and assumptions. The authors admit that we do see some in the originally submitted manuscript and did not present some data well. We now discuss our limitations in the revised manuscript.

Our biggest limitation in this study is that we do not have size-dependent chemical composition measurements (addressed above). Therefore, we will not be able to explain why INP varies from an aerosol chemistry perspective completely in this measurement report while we find some correlations or links with these factors. Indeed, more detailed process-level studies and/or closure studies are required to understand why one factor is more important than other.

The limitation regarding PSAP is addressed below in the specific question section.

It is also possible that due to the rarity of high-temperature INPs and the lower aerosol load at ENA, these INPs were present at levels below the detection limit of PINE-3 or a WT-CRAFT system. We did time-averaging the PINE-3 data to mitigate this issue.

The observed discrepancy between PINE-3 and WT-CRAFT data is likely due to (1) the aerosol sampling efficiency on filters or (2) a property inherent to the aerosol particle population at ENA, although elucidation of this property is difficult with current data.

(1) the sampling flow generally decreased over time, and the relative sampling flow deviation measured at the beginning and end of each sampling was on average (± standard error) larger at ENA (27.9 ± 3.0%) than that at SGP (5.1 ± 2.3%), which might hint to a decreased sampling efficiency at ENA towards the end of the sampling period. The larger deviation observed at ENA than at SGP is also presumably due to longer sampling intervals (typically ≈ 3 days; see Table S5) as compared to SGP (≈ 2 days; see Table S6), but the overall impact on $n_{INP}(T)$ deviation is not yet known.
(2) One possibility is that, although storage time in a -20 °C freezer might not have impacted $n_{INP}$ measured from filters substantially (Beall et al., 2020), any degradation of ice-active material occurred uniformly across the filter samples before the filters reached the storage site several thousand kilometers from the sampling site at ENA.

We clarified our points in Sect. 3.4 (L596-).

3. The analysis presented appears to be superficial and does not delve deeply into the complexities of the data. A more rigorous and detailed statistical analysis is crucial to understand the nuances and implications of your findings fully.

AR: The authors admit that INP data was not vigorously discussed previously. Our >45-day INP data is unique, and we took RC2's comment as guidance to improve the representativeness of our data in this measurement report. The authors have performed rigorous analyses of INP concentrations and freezing efficiencies, such as ice nucleation active surface site density, in comparison to previous and reference INP measurements (please see Sect. 3.2). We have also conducted the PINE-3 INP analyses based on different time-averaging intervals (6-hr, 12-hr, 24-hr, 48-hr, and 72-hr). A set of analysis data examples for SGP is shown in a series of figures below. The figures from ENA are available in the zipped files named: nINP_avg_plots_mar17 & nS_avg_plots_mar17.

The figures shown in the next page display histograms of the PINE-3 based $n_{INP}(T)$ Gaussian distribution with a degree temperature bin for a statistically validated freezing temperature range at SGP (-15 to -31 °C). Except for the first panel displaying the data of original measurement time resolution, the data were time-averaged for 6-hr, 12-hr, 24-hr, 48-hr, and 72-hr. As discussed in Sect. 3.2, we found reasonable data density (n > 77) across the measured freezing temperatures with a 6-hr time average. The numbers of data density across the assessed freezing temperatures for different time averages are available in the relative frequency plots for both $n_{INP}$ and $n_s$ in the zipped files.

[Figure]

[Figure]

Figures seen on the left show box plots of the PINE-3 based $n_{INP}(T)$ spectra with a degree temperature bin for a statistically validated freezing temperature range at SGP (-15 to -31 °C). Except for the first panel displaying the data of original measurement time resolution, the data were time-averaged for 6-hr, 12-hr, 24-hr, 48-hr, and 72-hr. Individual boxes display median (orange line), average (green line), 25% & 75% percentile (whiskers), and outliers beyond 5% & 95% percentile values.

Histograms seen in the next page represent the PINE-3 based $n_s(T)$ Gaussian distribution with a degree temperature bin for a statistically validated freezing temperature range at SGP (-15 to -31 °C). Except for the first panel displaying the data of original measurement time resolution, the data were time-averaged for 6-hr, 12-hr, 24-hr, 48-hr, and 72-hr.

[Figure]

[Figure]

Box plots on the left show the PINE-3 based $n_s(T)$ spectra with a degree temperature bin for a statistically validated freezing temperature range at SGP (-15 to -31 °C). Except for the first panel displaying the data of original measurement time resolution, the data were time-averaged for 6-hr, 12-hr, 24-hr, 48-hr, and 72-hr. Individual boxes display median (orange line), average (green line), 25% & 75% percentile (whiskers), and outliers beyond 5% & 95% percentile values.

The figures below show the $n_s(T)$ exponential fits for different time-averaged data from SGP and ENA. Following Li et al. (2022), we offer $n_s(T)$ parameterizations that fit the median values of the $n_s(T)$ data in individual one degree temperature bins. For both SGP and ENA, we found the highest correlation coefficients across the measured freezing temperatures with a 6-hr time average.

[Figure]

[Figure]

[Figure]

[Figure]

[Figure]

[Figure]

[Figure]

[Figure]

[Figure]

[Figure]

Based on our analyses, we identified that the lower detection of $n_{INP}$ ($\approx 0.03$ L$^{-1}$) with reasonable data density can be demonstrated by 6-hr time-averaged data. We also think 6-hr is representative of the time-averaged PINE-3 data because it includes the data from at least three complete temperature cycles and it provides a reasonable number of data points to be used for the synoptic scale analysis in the future. In the revised manuscript, we present our data with statistical context using box plots and histograms hinted by Li et al. (2022). Our updates are summarized in Sects. 3.2 and 3.3, as well as in Figures 5 and 6 (seen in the next page), in the revised manuscript. The authors also address how INP numbers vary with time as a function of wind properties (Sect. 3.6), back trajectory (Sect. 3.6), rain events (Sect. 3.2), and heat sensitivity (Sect. 3.3) besides the comparison over two sites.

[Figure]

Figure 5: Box plot of the PINE-3 based $n_{INP}(T)$ and $n_s(T)$ spectra with a degree temperature bin for a statistically validated freezing temperature range at SGP (A-B; -15 to -31 °C) and ENA (C-D; -20 to -31 °C). Individual boxes display median (orange line), average (green line), 25% & 75% percentile (whiskers), and outliers beyond 5% & 95% percentile values. The grey-shaded area in Panels A and C shows the maximum and minimum $n_{INP}(T)$ measured by DeMott et al. (2015) for SGP and previous INP studies from marine dominant sites located in the Atlantic Ocean for ENA (see Wilbourn et al. (2023) for more information). The reference $n_s(T)$ spectra in Panels B and D are adopted from U17 (Desert Dust, -14 to -30 °C, Ullrich et al., 2017), M18 (Sea Spray Aerosol, -20 to -28 °C, McCluskey et al., 2018)), H15a (illite NX, <-18 °C, Hiranuma et al., 2015a), and H15b as well as H19 (MCC, <-15 °C, Hiranuma et al., 2015b and 2019).

[Figure]

Figure 6: Histogram of the PINE-3 based $n_{INP}(T)$ and $n_s(T)$ Gaussian distribution with a degree temperature bin for a statistically validated freezing temperature range at SGP (A-B; -15 to -31 °C) and ENA (C-D; -20 to -31 °C). Individual data densities (# in red) and relative frequencies (Arbitrary Unit) for each degree are shown in each panel.

**Specific comments:**

1. In the Introduction, mischaracterize immersion freezing mechanisms INPs. The immersion freezing should also include the condensation freezing.

AR: L67-72 – The reviewer is correct that immersion freezing is also now called condensation freezing. We revised the description of immersion freezing that now includes condensation freezing. We rephrased L67-72 to "INPs can alter the altitude of ice cloud formation and influence nucleation and freezing pathways (Hoose and Möhler, 2012). For instance, water droplets containing these aerosol particles freeze at temperatures higher than would be possible with pure water alone. Especially, immersion freezing processes including a minor contribution of condensation freezing predominate over 85% of atmospheric heterogeneous freezing (Hande and Hoose, 2017; Westbrook and Illingworth, 2011)."

2. Regarding the comparison between PINE and offline freezing droplet measurements, it appears that no direct comparison has been provided. From the current presentation in Figure 6 and Figure 8, at least, it is not discernible. These figures should be redrawn to clearly illustrate the comparison and provide a more direct and insightful analysis of the results.

AR: Figure 8 & L595- The comparison is now revised in Sect. 3.4 with a new figure.

[Figure]

The figure above shows the comparison of online vs. offline $n_{INP}(T)$ spectra with blue and red box plots, respectively, from SGP (A) and ENA (B). The online PINE-3 spectra from SGP and ENA are time-averaged for 48-hour and 72-hour, respectively, to match with the aerosol sampling time intervals for the offline INP analysis. The offline spectra are all non-heated data. The structural description of individual boxes and grey-shaded areas are given in the Fig. 5 caption. Individual boxes display median (orange line), average (green line), 25% & 75% percentile (whiskers), and outliers beyond 5% & 95% percentile values. The grey-shaded area shows the maximum and minimum $n_{INP}(T)$ measured by DeMott et al. (2015) for SGP and previous INP studies from marine dominant sites located in the Atlantic Ocean for ENA (see Wilbourn et al. (2023) for more information).

3. About BC measurement using PASP, 'with the absorption at 529 nm most 330 representative of black carbon aerosols'. The PASP can provide aerosol absorption coefficients at three wavelength, the red light can represent BC characteristic more.

AR: L346-352 – The authors admit that our statement was misleading. We rephrased this part to: "Measured mass absorption cross-section values for freshly generated black carbon fall within a relatively narrow range of $7.5 \pm 1.2$ m$^2$g$^{-1}$ at 550 nm (Bond et al., 2013). This assumption of uniform aerosol composition may introduce uncertainties in information derived from PSAP data, which represents one of the limitations of this study, as few natural aerosol populations have uniform composition. In this study, the mass of black carbon present in aerosols ($m_{BC}$, ng m$^{-3}$) was estimated by dividing the absorption at 529 nm by the estimated mass absorbing cross-section of 7.5 m$^2$ g$^{-1}$ (Zheng et al., 2018; Bond et al., 2013)."

Bond et al. (2013) mention that the validity of BC's mass absorption cross-section measurement highly depends on the internal mixing state with other aerosol chemical components, which represents a notable limitation. For that matter, with long-term ground-level data, the authors think it might be important to study in the future what ice nucleation pathway is the most sensitive to the chemical mixing state of ambient aerosol particles. It is also important to examine if immersion and/or condensation freezing, requiring a water saturation condition or cloud condensation nuclei activation prior to ice nucleation, is the more predominant ice nucleation mechanism at the ARM mega sites.

**Citation**: https://doi.org/10.5194/egusphere-2023-1456-RC2

References

[revised manuscript text omitted]